# Towards Label Position Bias
# in Graph Neural Networks

**Haoyu Han[1], Xiaorui Liu[2], Feng Shi[3],**
**MohamadAli Torkamani[4]\*, Charu C. Aggarwal[5], Jiliang Tang[1]**
[1]Michigan State University    [2]North Carolina State University    [3]TigerGraph
[4] Amazon    [5]IBM T.J. Watson Research Center
{hanhaoy1,tangjili}@msu.edu,    xliu96@ncsu.edu
bill.shi@tigergraph.com,    alitor@amazon.com,    charu@us.ibm.com

## Abstract

Graph Neural Networks (GNNs) have emerged as a powerful tool for semi-supervised node classification tasks. However, recent studies have revealed various biases in GNNs stemming from both node features and graph topology. In this work, we uncover a new bias - label position bias, which indicates that the node closer to the labeled nodes tends to perform better. We introduce a new metric, the Label Proximity Score, to quantify this bias, and find that it is closely related to performance disparities. To address the label position bias, we propose a novel optimization framework for learning a label position unbiased graph structure, which can be applied to existing GNNs. Extensive experiments demonstrate that our proposed method not only outperforms backbone methods but also significantly mitigates the issue of label position bias in GNNs.

## 1  Introduction

Graph is a foundational data structure, denoting pairwise relationships between entities. It finds applications across a range of domains, such as social networks, transportation, and biology. [1, 2] Among these diverse applications, semi-supervised node classification has emerged as a crucial and challenging task, attracting significant attention from researchers. Given the graph structure, node features, and a subset of labels, the semi-supervised node classification task aims to predict the labels of unlabeled nodes. In recent years, Graph Neural Networks (GNNs) have demonstrated remarkable success in addressing this task due to their exceptional ability to model both the graph structure and node features [3]. A typical GNN model usually follows the message-passing scheme [4], which mainly contains two operators, i.e., feature transformation and feature propagation, to exploit node features, graph structure, and label information.

Despite the great success, recent studies have shown that GNNs could introduce various biases from the perspectives of node features and graph topology. In terms of node features, Jiang et al. [5] demonstrated that the message-passing scheme could amplify sensitive node attribute bias. A series of studies [6, 7, 8] have endeavored to mitigate this sensitive attribute bias in GNNs and ensure fair classification. In terms of graph topology, Tang et al. [9] investigated the degree bias in GNNs, signifying that high-degree nodes typically outperform low-degree nodes. This degree bias has also been addressed by several recent studies [10, 11, 12].

In addition to node features and graph topology, the label information, especially the position of labeled nodes, also plays a crucial role in GNNs. However, the potential bias in label information has been largely overlooked. In practice, with an equal number of training nodes, different labeling can result in significant discrepancies in test performance [13, 14, 15]. For instance, Ma et al. [16] study

---

*This work does not relate to the author's position at Amazon

37th Conference on Neural Information Processing Systems (NeurIPS 2023).

the subgroup generalization of GNNs and find that the shortest path distance to labeled nodes can also affect the GNNs' performance, but they haven't provided deep understanding or solutions. The investigation of the influence of labeled nodes' position on unlabeled nodes remains under-explored.

In this work, we discover the presence of a new bias in GNNs, namely the label position bias, which indicates that the nodes "closer" to the labeled nodes tend to receive better prediction accuracy. We propose a novel metric called Label Proximity Score (LPS) to quantify and measure this bias. Our study shows that different node groups with varied LPSs can result in a significant performance gap, which showcases the existence of label position bias. More importantly, this new metric has a much stronger correlation with performance disparity than existing metrics such as degree [9] and shortest path distance [16], which suggests that the proposed Label Proximity Score might be a more intrinsic measurement of label position bias.

Addressing the label position bias in GNNs is greatly desired. First, the label position bias would cause the fairness issue to nodes that are distant from the labeled nodes. For instance, in a financial system, label position bias could result in unfair assessments for individuals far from labeled ones, potentially denying them access to financial resources. Second, mitigating this bias has the potential to enhance the performance of GNNs, especially if nodes that are distant can be correctly classified. In this work, we propose a Label Position unbiased Structure Learning method (LPSL) to derive a graph structure that mitigates the label position bias. Specifically, our goal is to learn a new graph structure in which each node exhibits similar Label Proximity Scores. The learned graph structure can then be applied across various GNNs. Extensive experiments demonstrate that our proposed LPSL not only outperforms backbone methods but also significantly mitigates the issue of label position bias in GNNs.

## 2   Label Position Bias

In this section, we provide an insightful preliminary study to reveal the existence of label position bias in GNNs. Before that, we first define the notations used in this paper.

**Notations**. We use bold upper-case letters such as $\mathbf{X}$ to denote matrices. $\mathbf{X}_i$ denotes its $i$-th row and $\mathbf{X}_{ij}$ indicates the $i$-th row and $j$-th column element. We use bold lower-case letters such as $\mathbf{x}$ to denote vectors. $\mathbf{1_n} \in \mathbb{R}^{n \times 1}$ is all-ones column vector. The Frobenius norm and the trace of a matrix $\mathbf{X}$ are defined as $\|\mathbf{X}\|_F = \sqrt{\sum_{ij} \mathbf{X}_{ij}^2}$ and $tr(\mathbf{X}) = \sum_i \mathbf{X}_{ii}$, respectively. Let $\mathcal{G} = (\mathcal{V}, \mathcal{E})$ be a graph, where $\mathcal{V}$ is the node set and $\mathcal{E}$ is the edge set. $\mathcal{N}_i$ denotes the neighborhood node set for node $v_i$. The graph can be represented by an adjacency matrix $\mathbf{A} \in \mathbb{R}^{n \times n}$, where $\mathbf{A}_{ij} > 0$ indices that there exists an edge between nodes $v_i$ and $v_j$ in $\mathcal{G}$, or otherwise $\mathbf{A}_{ij} = 0$. Let $\mathbf{D} = diag(d_1, d_2, \ldots, d_n)$ be the degree matrix, where $d_i = \sum_j \mathbf{A}_{ij}$ is the degree of node $v_i$. The graph Laplacian matrix is defined as $\mathbf{L} = \mathbf{D} - \mathbf{A}$. We define the normalized adjacency matrix as $\tilde{\mathbf{A}} = \mathbf{D}^{-\frac{1}{2}} \mathbf{A} \mathbf{D}^{-\frac{1}{2}}$ and the normalized Laplacian matrix as $\tilde{\mathbf{L}} = \mathbf{I} - \tilde{\mathbf{A}}$. Furthermore, suppose that each node is associated with a $d$-dimensional feature $\mathbf{x}$ and we use $\mathbf{X} = [\mathbf{x}_1, \ldots, \mathbf{x}_n]^\top \in \mathrm{R}^{n \times d}$ to denote the feature matrix. In this work, we focus on the node classification task on graphs. Given a graph $\mathcal{G} = \{\mathbf{A}, \mathbf{X}\}$ and a partial set of labels $\mathcal{Y}_L = \{\mathbf{y}_1, \ldots, \mathbf{y}_l\}$ for node set $\mathcal{V}_L = \{v_1, \ldots, v_l\}$, where $\mathbf{y}_i \in \mathbb{R}^c$ is a one-hot vector with $c$ classes, our goal is to predict labels of unlabeled nodes. For convenience, we reorder the index of nodes and use a mask matrix $\mathbf{T} = \begin{bmatrix} \mathbf{I}_l & 0 \\ 0 & 0 \end{bmatrix}$ to represent the indices of labeled nodes.

**Label Proximity Score.** In this study, we aim to study the bias caused by label positions. When studying prediction bias, we first need to define the sensitive groups based on certain attributes or metrics. Therefore, we propose a novel metric, namely the Label Proximity Score, to quantify the closeness between test nodes and training nodes with label information. Specifically, the proposed Label Proximity Score (LPS) is defined as follows:

$$LPS = \mathbf{PT1_n}, \text{ and } \mathbf{P} = \left(\mathbf{I} - (1 - \alpha)\tilde{\mathbf{A}}\right)^{-1}, \tag{1}$$

where $\mathbf{P}$ represents the Personalized PageRank matrix, $\mathbf{T}$ is the label mask matrix, $\mathbf{1_n}$ is an all-ones column vector, and $\alpha \in (0, 1]$ stands for the teleport probability. $\mathbf{P}_{ij}$ represents the pairwise node proximity between node $i$ and node $j$. For each test node $i$, its LPS represents the sum of its node proximity values to all labeled nodes, i.e., $(\mathbf{PT1}_n)_i = \mathbf{P}_{i,:}\mathbf{T1}_n = \sum_{j \in \mathcal{V}_L} \mathbf{P}_{ij}$.

**Sensitive Groups.** In addition to the proposed LPS, we also explore two existing metrics such as node degree [9] and shortest path distance to label nodes [16] for comparison since they could be related to the label position bias. For instance, the node with a high degree is more likely to connect with labeled nodes, and the node with a small shortest path to a labeled node is also likely "closer" to all labeled nodes if the number of labeled nodes is small. According to these metrics, we split test nodes into different sensitive groups. Specifically, for node degree and shortest path distance to label nodes, we use their actual values to split them into seven sensitive groups, as there are only very few nodes whose degrees or shortest path distances are larger than seven. For the proposed LPS, we first calculate its value and subsequently partition the test nodes evenly into seven sensitive groups, each having an identical range of LPS values.

**Experimental Setup**. We conduct the experiments on three representative datasets used in semi-supervised node classification tasks, namely Cora, CiteSeer, and PubMed. We also experiment with three different labeling rates: 5 labels per class, 20 labels per class, and 60% labels per class. The experiments are performed using two representative GNN models, GCN [17] and APPNP [18], which cover both coupled and decoupled architectures. We also provide the evaluation on Label Propagation (LP) [19] to exclude the potential bias caused by node features. For GCN and APPNP, we adopt the same hyperparameter setting with their original papers. The node classification accuracy on different sensitive groups $\{1, 2, 3, 4, 5, 6, 7\}$ with the labeling rate of 20 labeled nodes per class under APPNP, GCN, and LP models is illustrated in Figure 1, 2, and 3 respectively. Due to the space limitation, we put more details and results of other models, datasets, and labeling rates into Appendix A.

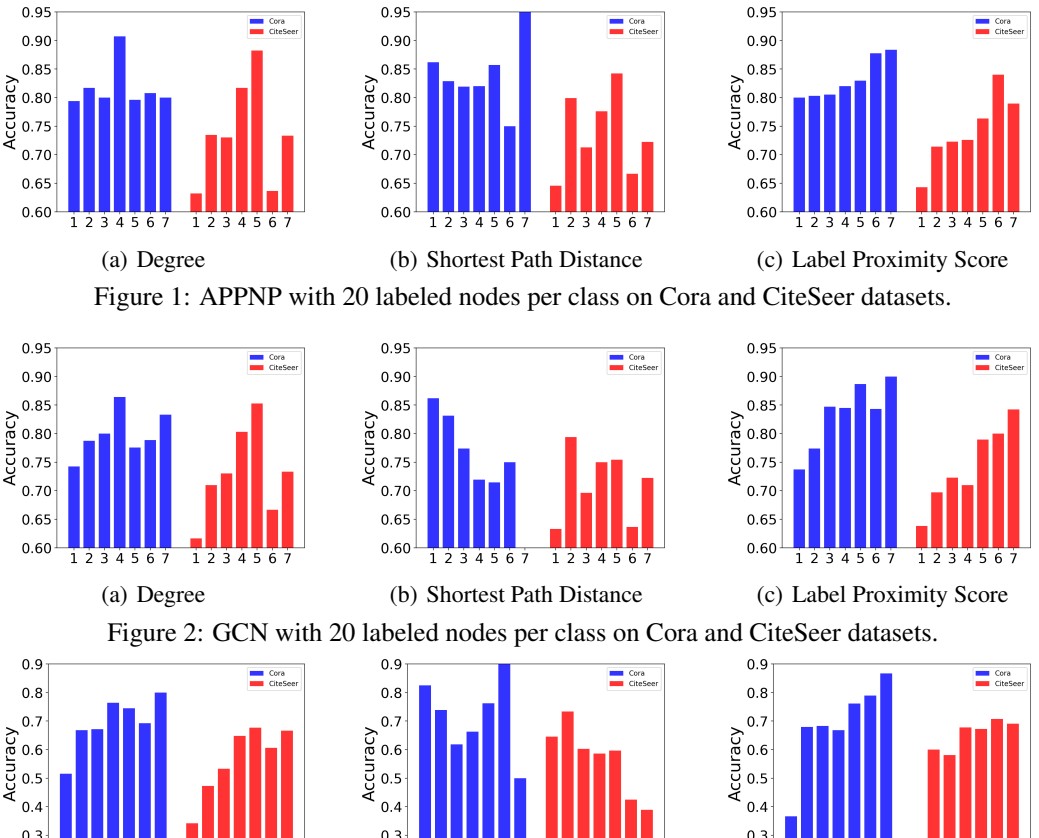

(a) Degree      (b) Shortest Path Distance      (c) Label Proximity Score

Figure 1: APPNP with 20 labeled nodes per class on Cora and CiteSeer datasets.

(a) Degree      (b) Shortest Path Distance      (c) Label Proximity Score

Figure 2: GCN with 20 labeled nodes per class on Cora and CiteSeer datasets.

(a) Degree      (b) Shortest Path Distance      (c) Label Proximity Score

Figure 3: LP with 20 labeled nodes per class on Cora and CiteSeer datasets.

**Observations.** From the results presented in Figure 1, 2, and 3, we can observe the following:

- Label Position bias is prevalent across all GNN models and datasets. The classification accuracy can notably vary between different sensitive groups, and certain trends are discernible. To

ensure fairness and improve performance, addressing this bias is a crucial step in improving GNN models.

- While Degree and Shortest Path Distance (SPD) can somewhat reflect disparate performance, indicating that nodes with higher degrees and shorter SPDs tend to perform better, these trends lack consistency, and they can't fully reflect the Label Position bias. For instance, degree bias is not pronounced in the APPNP model as shown in Figure 1, as APPNP can capture the global structure. Moreover, SPD fails to effectively evaluate relatively low homophily graphs, such as CiteSeer [20]. Consequently, there is a need to identify a more reliable metric.

- The Label Proximity Score (LPS) consistently exhibits a strong correlation with performance disparity across all datasets and models. Typically, nodes with higher LPS scores perform better. In addition, nodes with high degrees and low Shortest Path Distance (SPD) often have higher LPS, as previously analyzed. Therefore, LPS is highly correlated with label position bias.

- The Label Propagation, which solely relies on the graph structure, demonstrates a stronger label position bias compared to GNNs as shown in Figure 3. Moreover, the label position bias becomes less noticeable in all models when the labeling rate is high, as there typically exist labeled nodes within the two-hop neighborhood of each test node (detailed in Appendix A). These observations suggest that the label position bias is predominantly influenced by the graph structure. Consequently, this insight motivates us to address the Label Position bias from the perspective of the graph structure.

In conclusion, label position bias is indeed present in GNN models, and the proposed Label Proximity Score accurately and consistently reflects the performance disparity over different sensitive groups for different models across various datasets. Overall, the label proximity score exhibits more consistent and stronger correlations with performance disparity compared with node degree and shortest path distance, which suggests that LPS serves as a better metric for label position bias. Further, through the analysis of the Label Propagation method and the effects of different labeling rates, we deduce that the label position bias is primarily influenced by the graph structure. This understanding paves us a way to mitigate label position bias.

## 3 The Proposed Framework

The studies in Section 2 suggest that Label Position bias is a prevalent issue in GNNs. In other words, nodes far away from labeled nodes tend to yield subpar performance. Such unfairness could be problematic, especially in real-world applications where decisions based on these predictions can have substantial implications. As a result, mitigating label position bias has the potential to enhance the fairness of GNNs in real-world applications, as well as improve overall model performance. Typically, there are two ways to address this problem, i.e., from a model-centric or a data-centric perspective. In this work, we opt for a data-centric perspective for two primary reasons: (1) The wide variety of GNN models in use in real-world scenarios, each with its unique architecture, makes it challenging to design a universal component that can be seamlessly integrated into all GNNs to mitigate the label position bias. Instead, the graph structure is universal and can be applied to any existing GNNs. (2) Our preliminary studies indicate that the graph structure is the primary factor contributing to the label position bias. Therefore, it is more rational to address the bias by learning a label position unbiased graph structure.

However, there are mainly two challenges: (1) How can we define a label position unbiased graph structure, and how can we learn this structure based on the original graph? (2) Given that existing graphs are typically sparse, how can we ensure that the learned data structure is also sparse to avoid excessive memory consumption? In the following subsections, we aim to address these challenges.

### 3.1 Label Position Unbiased Graph Structure Learning

Based on our preliminary studies, the Label Proximity Score (LPS) can consistently reflect performance disparity across various GNNs and indicate the label position bias. Therefore, to mitigate the label position bias from the structural perspective, our objective is to learn a new graph structure in which each node exhibits similar LPSs. Meanwhile, this learned unbiased graph structure should maintain certain properties of the original graph. To achieve this goal, we formulate the Label Position

Unbiased Structure Learning (LPSL) problem as follows:

$$\underset{\mathbf{B}}{\arg\min} \|\mathbf{I} - \mathbf{B}\|_F^2 + \lambda \text{tr}(\mathbf{B}^\top \tilde{\mathbf{L}} \mathbf{B})$$
$$\text{s.t.} \quad \mathbf{BT1}_n = c\mathbf{1}_n, \mathbf{B}_{ij} \geq 0 \quad \forall i,j \tag{2}$$

where $\mathbf{B} \in \mathbb{R}^{n \times n}$ represents the debiased graph structure matrix. $\text{tr}(\mathbf{B}^\top \tilde{\mathbf{L}} \mathbf{B}) = \sum_{(v_i, v_j) \in \mathcal{E}} \|\mathbf{B}_i / \sqrt{d_i} - \mathbf{B}_j / \sqrt{d_j}\|_2^2$ measures the smoothness of the new structure based on the original graph structure. The proximity to identity matrix $\mathbf{I} \in \mathbb{R}^{n \times n}$ encourages self-loops and avoids trivial over-smoothed structures. $\lambda$ is a hyperparameter that controls the balance between smoothness and self-loop. $\mathbf{T}$ is the mask matrix indicating the labeled nodes, $\mathbf{1_n}$ is the all-ones vector, and $c$ is a hyperparameter serving as the uniform Label Proximity Score for all nodes. Due to the $\mathbf{B}$ represents the graph structure, we have the constraint that all elements in $\mathbf{B}$ should be non-negative.

Notably, if we ignore the constraint, then the optimal solution for this primary problem is given by $\mathbf{B} = (\mathbf{I} + \lambda \mathbf{L})^{-1} = \alpha(\mathbf{I} - (1 - \alpha \tilde{\mathbf{A}}))^{-1}$, where $\alpha = \frac{1}{1+\lambda}$. This solution recovers the Personalized PageRank (PPR) matrix which measures pairwise node proximity. Furthermore, the constraint in Eq. (2) ensures that all nodes have the same Label Proximity Score, denoted as $c$. The constraint encourages fair label proximity scores for all nodes so that the learned graph structure mitigates the label position bias.

The constrained optimization problem in Eq. (2) is a convex optimization problem, and it can be solved by the Lagrange Multiplier method with the projected gradient descent [21]. The augmented Lagrange function can be written as:

$$L_\rho(\mathbf{B}, \mathbf{y}) = \|\mathbf{I} - \mathbf{B}\|_F^2 + \lambda \text{tr}(\mathbf{B}^\top \tilde{\mathbf{L}} \mathbf{B}) + \mathbf{y}^\top(\mathbf{BT1}_n - c\mathbf{1}_n) + \frac{\rho}{2}\|\mathbf{BT1}_n - c\mathbf{1}_n\|_2^2, \tag{3}$$

where $\mathbf{y} \in \mathbb{R}^{n \times 1}$ is the introduced Lagrange multiplier, and $\rho > 0$ represents the augmented Lagrangian parameter. The gradient of $L_\rho(\mathbf{B}, \mathbf{y})$ to $\mathbf{B}$ can be represented as:

$$\frac{\partial L_\rho}{\partial \mathbf{B}} = 2(\mathbf{B} - \mathbf{I}) + 2\lambda \tilde{\mathbf{L}} \mathbf{B} + \mathbf{y}(\mathbf{T1}_n)^\top + \rho(\mathbf{BT1}_n - c\mathbf{1}_n)(\mathbf{T1}_n)^\top. \tag{4}$$

Then, the problem can be solved by dual ascent algorithm [22] as follows:

$$\mathbf{B}^{k+1} = \underset{\mathbf{B}}{\arg\min} L_\rho(\mathbf{B}^k, \mathbf{y}^k)$$
$$\mathbf{B}_{ij}^{k+1} = \max(0, \mathbf{B}_{ij}^{k+1})$$
$$\mathbf{y}^{k+1} = \mathbf{y}^k + \rho(\mathbf{B}^k \mathbf{T1}_n - c\mathbf{1}_n),$$

where $k$ is the current optimization step, and $\mathbf{B}^{k+1}$ can be obtained by multiple steps of gradient descent using the gradient in Eq. (4).

## 3.2 Understandings

In this subsection, we provide the understanding and interpretation of our proposed LPSL, establishing its connections with the message passing in GNNs.

*Remark* 3.1. The feature aggregation using the learned graph structure $\mathbf{B}$ directly as a propagation matrix, i.e., $\mathbf{F} = \mathbf{BX}$, is equivalent to applying the message passing in GNNs using the original graph if $\mathbf{B}$ is the approximate or exact solution to the primary problem defined in Eq. 2 without constraints.

The detailed proof can be found in Appendix B. Remark 3.1 suggests that we can directly substitute the propagation matrix in GNNs with the learned structure $\mathbf{B}$. The GNNs are trained based on the labeled nodes, and the labeled nodes would influence the prediction of unlabeled nodes because of the message-passing scheme. Following the definition in [23], the influence of node $j$ on node $i$ can be represented by $I_i(j) = sum\left[\frac{\partial \mathbf{h}_i}{\partial \mathbf{x}_j}\right]$, where $\mathbf{h}_i$ is the representation of node $i$, $\mathbf{x}_j$ is the input feature of node $j$, and $\left[\frac{\partial \mathbf{h}_i}{\partial \mathbf{x}_j}\right]$ represents the Jacobian matrix. Afterward, we have the following Proposition based on the influence scores:

**Proposition 3.1.** *The influence scores from all labeled nodes to any unlabeled node $i$ will be the equal, i.e., $\sum_{j \in \mathcal{V}_L} I_i(j) = c$, when using the unbiased graph structure $\mathbf{B}$ obtained from the optimization problem in Eq. (2) as the propagation matrix in GNNs.*

The proof can be found in Appendix B. Proposition 3.1 suggests that by using the unbiased graph structure for feature propagation, each node can receive an equivalent influence from all the labeled nodes, thereby mitigating the label position bias issue.

### 3.3 $\ell_1$-regularized Label Position Unbiased Sparse Structure Learning

One challenge of solving the graph structure learning problem in Eq. (2) is that it could result in a dense structure matrix $\mathbf{B} \in \mathbb{R}^{n \times n}$. This is a memory-intensive outcome, especially when the number of nodes $n$ is large. Furthermore, applying this dense matrix to GNNs can be time-consuming for downstream tasks, which makes it less practical for real-world applications. To make the learned graph structure sparse, we propose the following $\ell_1$-regularized Label Position Unbiased Sparse Structure Learning optimization problem:

$$\arg\min_B \|\mathbf{I} - \mathbf{B}\|_F^2 + \lambda \text{tr}(\mathbf{B}^\top \mathbf{L} \mathbf{B}) + \beta \|\mathbf{B}\|_1$$
$$\text{s.t.} \quad \mathbf{B}\mathbf{T}\mathbf{1}_n = c\mathbf{1}_n, \mathbf{B}_{ij} \geq 0 \quad \forall i, j,$$
(5)

where $\|\mathbf{B}\|_1$ represents the $\ell_1$ regularization that encourages zero values in $\mathbf{B}$. $\beta > 0$ is a hyper-parameter to control the sparsity of $\mathbf{B}$. The primary problem in Eq. (5) is proved to have a strong localization property and can guarantee the sparsity [24, 25]. The problem in Eq. (5) can also be solved by the Lagrange Multiplier method. However, when the number of nodes $n$ is large, solving this problem using conventional gradient descent methods becomes computationally challenging. Therefore, we propose to solve the problem in Eq. (5) efficiently by Block Coordinate Descent (BCD) method [26] in conjunction with the proximal gradient approach, particularly due to the presence of the $\ell_1$ regularization. Specifically, we split $\mathbf{B}$ into column blocks, and $\mathbf{B}_{:,j}$ represents the $j$-th block. The gradient of $L_\rho$ with respect to $\mathbf{B}_{:,j}$ can be written as:

$$\frac{\partial L_\rho}{\partial \mathbf{B}_{:,j}} = 2(\mathbf{B}_{:,j} - \mathbf{I}_{:,j}) + 2\lambda \tilde{\mathbf{L}} \mathbf{B}_{:,j} + \mathbf{y}(\mathbf{T}\mathbf{1}_n)_j^\top + \rho(\mathbf{B}\mathbf{T}\mathbf{1}_n - c\mathbf{1}_n)(\mathbf{T}\mathbf{1}_n)_j^\top,$$
(6)

where $(\mathbf{T}\mathbf{1}_n)_j \in \mathbb{R}^{d \times 1}$ is the corresponding block part with block size $d$. After updating the current block $\mathbf{B}_{:,j}$, we apply a soft thresholding operator $S_{\beta/\rho}(\cdot)$ based on the proximal mapping. The full algorithm is detailed in Algorithm 1. Notably, lines 6-8 handle the block updates, line 9 performs the soft thresholding operation, and line 11 updates Lagrange multiplier $\mathbf{y}$ through dual ascent update.

### 3.4 The Model Architecture

The proposed LPSL learns an unbiased graph structure with respect to the labeled nodes. Therefore, the learned graph structure can be applied to various GNN models to mitigate the Label Position bias. In this work, we test LPSL on two widely used GNN models, i.e., GCN [17] and APPNP [18]. For the GCN model, each layer can be represented by:

$$\mathbf{H}^{l+1} = \sigma\left(\mathbf{B}_\lambda \mathbf{H}^l \mathbf{W}^l\right),$$

where $\mathbf{H}^0 = \mathbf{X}$, $\sigma$ is the non-linear activation function, $\mathbf{B}_\lambda$ is the unbiased structure with papermeter $\lambda$, and $\mathbf{W}^l$ is the weight matrix in the $l$-th layer. We refer to this model as LPSL$_{\text{GCN}}$. For the APPNP model, we directly use the learned $\mathbf{B}_\lambda$ as the propagation matrix, and the prediction can be written as:

$$\mathbf{Y}_{\text{pred}} = \mathbf{B}_\lambda f_\theta(\mathbf{X}),$$

---

**Algorithm 1** Algorithm of LPSL

1: **Input:** Laplacian matrix $\tilde{\mathbf{L}}$, Label mask matrix $\mathbf{T}$, Hyperparamters $\lambda, c, \beta, \rho$, learning rate $\gamma$
2: **Output**: Label position unbiased graph structure $\mathbf{B}$

3: **Initialization**: $\mathbf{B}^0 = \mathbf{I}$ and $\mathbf{y}^0 = \mathbf{0}$
4: **while** Not converge **do**
5:     **for** each block $j$ **do**
6:         **for** $i = 0$ **to** update steps $t$ **do**
7:             $\mathbf{B}_{:,j} = \mathbf{B}_{:,j} - \gamma * \frac{\partial L_\rho}{\partial \mathbf{B}_{:,j}}$
8:         **end for**
9:         $\mathbf{B}_{:,j} = S_{\beta/\rho}(\mathbf{B}_{:,j})$
10:     **end for**
11:     $\mathbf{y} = \mathbf{y} + \rho(\mathbf{B}\mathbf{T}\mathbf{1}_n - c\mathbf{1}_n)$
12: **end while**

13: **return** $\mathbf{B}$

---

where $f_\theta(\cdot)$ is any machine learning model parameterized by the learnable parameters $\theta$. We name this model as LPSL$_{\text{APPNP}}$. The parameter $\lambda$ provides a high flexibility when applying $\mathbf{B}_\lambda$ to different

GNN architectures. For decoupled GNNs such as APPNP, which only propagates once, a large $\lambda$ is necessary to encode the global graph structure with a denser sparsity. In contrast, for coupled GNNs, such as GCN, which apply propagation multiple times, a smaller $\lambda$ can be used to encode a more local structure with a higher sparsity. Our code is available at: https://github.com/haoyuhan1/LPSL.

# 4 Experiment

In this section, we conduct comprehensive experiments to verify the effectiveness of the proposed LPSL. In particular, we try to answer the following questions:

- **Q1:** Can the proposed LPSL improve the performance of different GNNs? (Section 4.2)
- **Q2:** Can the proposed LPSL mitigate the label position bias? (Section 4.3)
- **Q3:** How do different hyperparameters affect the proposed LPSL? (Section 4.4)

## 4.1 Experimental Settings

**Datasets.** We conduct experiments on 8 real-world graph datasets for the semi-supervised node classification task, including three citation datasets, i.e., Cora, Citeseer, and Pubmed [27], two co-authorship datasets, i.e., Coauthor CS and Coauthor Physics, two co-purchase datasets, i.e., Amazon Computers and Amazon Photo [28], and one OGB dataset, i.e., ogbn-arxiv [29]. The details about these datasets are shown in Appendix C.

We employ the fixed data split for the ogbn-arxiv dataset, while using ten random data splits for all other datasets to ensure more reliable results [30]. Additionally, for the Cora, CiteSeer, and PubMed datasets, we experiment with various labeling rates: low labeling rates with 5, 10, and 20 labeled nodes per class, and high labeling rates with 60% labeled nodes per class. Each model is run three times for every data split, and we report the average performance along with the standard deviation.

**Baselines.** To the best of our knowledge, there are no previous works that aim to address the label position bias. In this work, we select three GNNs, namely, GCN [17], GAT [31], and APPNP [18], two Non-GNNs, MLP and Label Propagation [19], as baselines. Furthermore, we also include GRADE [32], a method designed to mitigate degree bias. Notably, SRGNN [33] demonstrates that if labeled nodes are gathered locally, it could lead to an issue of feature distribution shift. SRGNN aims to mitigate the feature distribution shift issue and is also included as a baseline.

**Hyperparameters Setting.** We follow the best hyperparameter settings in their original papers for all baselines. For the proposed LPSL$_{GCN}$, we set the $\lambda$ in range [1,8]. For LPSL$_{APPNP}$, we set the $\lambda$ in the range [8, 15]. For both methods, $c$ is set in the range [0.5, 1.5]. We fix the learning rate 0.01, dropout 0.5 or 0.8, hidden dimension size 64, and weight decay 0.0005, except for the ogbn-arxiv dataset. Adam optimizer [34] is used in all experiments. More details about the hyperparameters setting for all methods can be found in Appendix D.

## 4.2 Performance Comparison on Benchmark Datasets

In this subsection, we test the learned unbiased graph structure by the proposed LPSL on both GCN and APPNP models. We then compare these results with seven baseline methods across all eight datasets. The primary results are presented in Table 1. Due to space limitations, we have included the results from other baselines in Appendix E. From these results, we can make several key observations:

- The integration of our proposed LPSL to both GCN and APPNP models consistently improves their performance on almost all datasets. This indicates that a label position unbiased graph structure can significantly aid semi-supervised node classification tasks.
- Concerning the different labeling rates for the first three datasets, our proposed LPSL shows greater performance improvement with a low labeling rate. This aligns with our preliminary study that label position bias is more pronounced when the labeling rate is low.
- SRGNN, designed to address the feature distribution shift issue, does not perform well on most datasets with random splits instead of locally distributed labels. Only when the labeling rate is very low, SRGNN can outperform GCN. Hence, the label position bias cannot be solely solved by addressing the feature distribution shift.
- The GRADE method, aimed at mitigating the degree-bias issue, also fails to improve overall performance with randomly split datasets.

Table 1: Semi-supervised node classification accuracy (%) on benchmark datasets.

| Dataset | Label Rate | GCN | APPNP | GRADE | SRGNN | LPSL$_{GCN}$ | LPSL$_{APPNP}$ |
|---|---|---|---|---|---|---|---|
| Cora | 5 | 70.68 ± 2.17 | 75.86 ± 2.34 | 69.51 ± 6.79 | 70.77 ± 1.82 | 76.58 ± 2.37 | **77.24 ± 2.18** |
| | 10 | 76.50 ± 1.42 | 80.29 ± 1.00 | 74.95 ± 2.46 | 75.42 ± 1.57 | 80.39 ± 1.17 | **81.59 ± 0.98** |
| | 20 | 79.41 ± 1.30 | 82.34 ± 0.67 | 77.41 ± 1.49 | 78.42 ± 1.75 | 82.74 ± 1.01 | **83.24 ± 0.75** |
| | 60% | 88.60 ± 1.19 | 88.49 ± 1.28 | 86.84 ± 0.99 | 87.17 ± 0.95 | **88.75 ± 1.21** | 88.62 ± 1.69 |
| CiteSeer | 5 | 61.27 ± 3.85 | 63.92 ± 3.39 | 63.03 ± 3.61 | 64.84 ± 3.41 | 65.65 ± 2.47 | **65.70 ± 2.18** |
| | 10 | 66.28 ± 2.14 | 67.57 ± 2.05 | 64.20 ± 3.23 | 67.83 ± 2.19 | 67.73 ± 2.57 | **68.76 ± 1.77** |
| | 20 | 69.60 ± 1.67 | 70.85 ± 1.45 | 67.50 ± 1.76 | 69.13 ± 1.99 | 70.73 ± 1.32 | **71.25 ± 1.14** |
| | 60% | 76.88 ± 1.78 | 77.42 ± 1.47 | 74.00 ± 1.87 | 74.57 ± 1.57 | 77.18 ± 1.64 | **77.56 ± 1.44** |
| PubMed | 5 | 69.76 ± 6.46 | 72.68 ± 5.68 | 66.90 ± 6.49 | 69.38 ± 6.48 | 73.46 ± 4.64 | **73.57 ± 5.30** |
| | 10 | 72.79 ± 3.58 | 75.53 ± 3.85 | 73.31 ± 3.75 | 72.69 ± 3.49 | 75.67 ± 4.42 | **76.18 ± 4.05** |
| | 20 | 77.43 ± 2.66 | 78.93 ± 2.11 | 75.12 ± 2.37 | 77.09 ± 1.68 | 78.75 ± 2.45 | **79.26 ± 2.32** |
| | 60% | **88.48 ± 0.46** | 87.56 ± 0.52 | 86.90 ± 0.46 | 88.32 ± 0.55 | 87.75 ± 0.57 | 87.96 ± 0.57 |
| CS | 20 | 91.73 ± 0.49 | 92.38 ± 0.38 | 89.43 ± 0.67 | 89.43 ± 0.67 | 91.94 ± 0.54 | **92.44 ± 0.36** |
| Physics | 20 | 93.29 ± 0.80 | 93.49 ± 0.67 | 91.44 ± 1.41 | 93.16 ± 0.64 | 93.56 ± 0.51 | **93.65 ± 0.50** |
| Computers | 20 | 79.17 ± 1.92 | 79.07 ± 2.34 | 79.01 ± 2.36 | 78.54 ± 2.15 | **80.05 ± 2.92** | 79.58 ± 2.31 |
| Photo | 20 | 89.94 ± 1.22 | 90.87 ± 1.14 | 90.17 ± 0.93 | 89.36 ± 1.02 | 90.85 ± 1.16 | **90.93 ± 1.40** |
| ogbn-arxiv | 54% | 71.91 ± 0.15 | 71.61 ± 0.30 | OOM | 68.01 ± 0.35 | **72.04 ± 0.12** | 69.20 ± 0.26 |

## 4.3 Evaluating Bias Mitigation Performance

In this subsection, we aim to investigate whether the proposed LPSL can mitigate the label position bias. We employ all three aforementioned bias metrics, namely label proximity score, degree, and shortest path distance, on Cora and CiteSeer datasets. We first group test nodes into different sensitive groups according to the metrics, and then use three representative group bias measurements - Weighted Demographic Parity (WDP), Weighted Standard Deviation (WSD), and Weighted Coefficient of Variation (WCV) - to quantify the bias. These are defined as follows:

$$\text{WDP} = \frac{\sum_{i=1}^{D} N_i \cdot |A_i - A_{\text{avg}}|}{N_{\text{total}}}, \text{WSD} = \sqrt{\frac{1}{N_{\text{total}}} \sum_{i=1}^{D} N_i \cdot (A_i - A_{\text{avg}})^2}, \text{WCV} = \frac{\text{WSD}}{A_{\text{avg}}},$$

where $D$ is the number of groups, $N_i$ is the node number of group $i$, $A_i$ is the accuracy of group $i$, $A_{\text{avg}}$ is the weighted average accuracy of all groups, i.e., the overall accuracy, and $N_{\text{total}}$ is the total number of nodes. We choose six representative models, i.e., Label Propagation (LP), GRADE, GCN, APPNP, LPSL$_{GCN}$, and LPSL$_{APPNP}$, in this experiment. The results of the label proximity score, degree, and shortest path on the Cora and Citeseer datasets are shown in Tabel 2, 3, and 4, respectively. It can be observed from the tables:

- The Label Propagation method, which solely utilizes the graph structure information, exhibits the most significant label position bias across all measurements and datasets. This evidence suggests that label position bias primarily stems from the biased graph structure, thereby validating our strategy of learning an unbiased graph structure with LPSL.

- The proposed LPSL not only enhances the classification accuracy of the backbone models, but also alleviates the bias concerning Label Proximity Score, degree, and Shortest distance.

- The GRADE method, designed to mitigate degree bias, does exhibit a lesser degree bias than GCN and APPNP. However, it still falls short when compared to the proposed LPSL. Furthermore, GRADE may inadvertently heighten the bias evaluated by other metrics. For instance, it significantly increases the label proximity score bias on the CiteSeer dataset.

Table 2: Comparison of Methods in Addressing Label Proximity Score Bias.

| Dataset | Cora | | | CiteSeer | | |
|---|---|---|---|---|---|---|
| Method | WDP ↓ | WSD ↓ | WCV ↓ | WDP ↓ | WSD ↓ | WCV ↓ |
| LP | 0.1079 | 0.1378 | 0.1941 | 0.2282 | 0.2336 | 0.4692 |
| GRADE | 0.0372 | 0.0489 | 0.0615 | 0.0376 | 0.0467 | 0.0658 |
| GCN | 0.0494 | 0.0618 | 0.0758 | 0.0233 | 0.0376 | 0.0524 |
| LPSL$_{GCN}$ | **0.0361** | **0.0438** | **0.0518** | **0.0229** | **0.0346** | 0.0476 |
| APPNP | 0.0497 | 0.0616 | 0.0732 | 0.0344 | 0.0426 | 0.0594 |
| LPSL$_{APPNP}$ | 0.0390 | 0.0476 | 0.0562 | 0.0275 | 0.0349 | **0.0474** |

Table 3: Comparison of Methods in Addressing Degree Bias.

| Dataset | Cora | | | CiteSeer | | |
|---|---|---|---|---|---|---|
| Method | WDP ↓ | WSD ↓ | WCV ↓ | WDP ↓ | WSD ↓ | WCV ↓ |
| LP | 0.0893 | 0.1019 | 0.1447 | 0.1202 | 0.1367 | 0.2773 |
| GRADE | 0.0386 | 0.0471 | 0.0594 | 0.0342 | 0.0529 | 0.0744 |
| GCN | 0.0503 | 0.0566 | 0.0696 | 0.0466 | 0.0643 | 0.0901 |
| LPSL$_{\text{GCN}}$ | 0.0407 | 0.0468 | 0.0554 | 0.0378 | 0.0538 | 0.0742 |
| APPNP | 0.0408 | 0.0442 | 0.0527 | 0.0499 | 0.0688 | 0.0964 |
| LPSL$_{\text{APPNP}}$ | **0.0349** | **0.0395** | **0.0467** | **0.0316** | **0.0487** | **0.0665** |

Table 4: Comparison of Methods in Addressing Shortest Path Distance Bias.

| DataSet | Cora | | | CiteSeer | | |
|---|---|---|---|---|---|---|
| Method | WDP ↓ | WSD ↓ | WCV ↓ | WDP ↓ | WSD ↓ | WCV ↓ |
| LP | 0.0562 | 0.0632 | 0.0841 | 0.0508 | 0.0735 | 0.109 |
| GRADE | 0.0292 | 0.0369 | 0.0459 | 0.0282 | 0.0517 | 0.0707 |
| GCN | 0.0237 | 0.0444 | 0.0533 | 0.0296 | 0.0553 | 0.0752 |
| LPSL$_{\text{GCN}}$ | **0.0150** | **0.0248** | **0.0289** | **0.0246** | 0.0526 | 0.0714 |
| APPNP | 0.0218 | 0.0316 | 0.0369 | 0.0321 | 0.0495 | 0.0668 |
| LPSL$_{\text{APPNP}}$ | 0.0166 | 0.0253 | 0.0295 | 0.0265 | **0.0490** | **0.0654** |

## 4.4 Ablation Study

In this subsection, we first explore the impact of different hyperparameters, specifically the smoothing term $\lambda$ and the constraint $c$, on our model. We conducted experiments on the Cora and CiteSeer datasets using ten random data splits with 20 labels per class. The accuracy of different $\lambda$ values for LPSL$_{\text{APPNP}}$ and LPSL$_{\text{GCN}}$ on the Cora and CiteSeer datasets are illustrated in Figure 4.

From the results, we note that the proposed LPSL is not highly sensitive to the $\lambda$ within the selected regions. Moreover, for the APPNP model, the best $\lambda$ is higher than that for the GCN model, which aligns with our discussion in Section 3 that the decoupled GNNs require a larger $\lambda$ to encode the global graph structure. The results for hyperparameter $c$ can be found in Appendix F with similar observations.

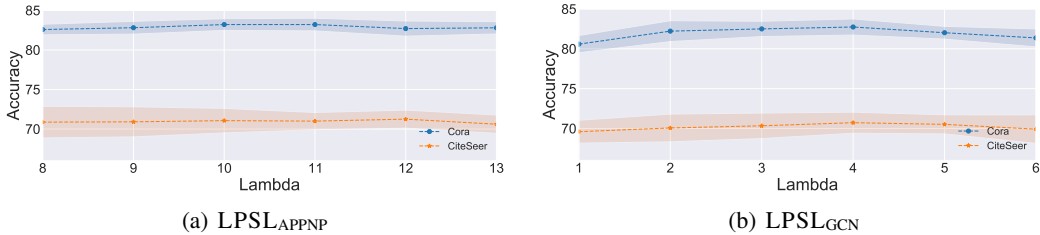

(a) LPSL$_{\text{APPNP}}$                 (b) LPSL$_{\text{GCN}}$

Figure 4: The accuracy of different $\lambda$ for LPSL$_{\text{APPNP}}$ and LPSL$_{\text{GCN}}$ on Cora and CiteSeer datasets.

In addition, we investigate the impact of the sparse graph structure matrix $\mathbf{B}$ generated using $l_1$ regularization, as described in Eq. 5, on the performance of different models. For this purpose, we utilize the Cora dataset and select an appropriate $\beta$ value to produce the sparse graph structure matrix $\mathbf{B}$ at sparsity levels of 80%, 90%, and 95%. This means that 80%, 90%, and 95% of the entries in the matrix $\mathbf{B}$ are zero, respectively.

The results are presented in Table 5. We observe that the accuracy at 80% and 90% sparsity is closely aligned with that of the dense matrix. However, when sparsity reaches 95%, there is a slight drop in accuracy, mirroring the findings of PPRGo [35]. Notably, fairness appears to be relatively unaffected by the varying levels of sparsity.

Table 5: Performance comparison using different sparsity levels of **B** on the Cora dataset.

| Sparsity | LPSL$_{\text{APPNP}}$ | | | | LPSL$_{\text{GCN}}$ | | | |
|---|---|---|---|---|---|---|---|---|
| | 15%($\beta = 0$) | 80% | 90% | 95% | 15%($\beta = 0$) | 80% | 90% | 95% |
| ACC | 83.24 | 83.20 | 82.67 | 81.90 | 82.74 | 82.66 | 82.70 | 81.78 |
| WDP | 0.033 | 0.0345 | 0.0323 | 0.0322 | 0.0347 | 0.0346 | 0.0347 | 0.0337 |
| WSD | 0.042 | 0.0444 | 0.0417 | 0.0411 | 0.0443 | 0.0439 | 0.0425 | 0.0433 |
| WCV | 0.0505 | 0.0534 | 0.0504 | 0.0503 | 0.0536 | 0.0532 | 0.0515 | 0.053 |

## 5 Related Work

Graph Neural Networks (GNNs) serve as an effective framework for representing graph-structured data, primarily employing two operators: feature transformation and propagation. The ordering of these operators classifies most GNNs into two categories: Coupled and Decoupled GNNs. Coupled GNNs, such as GCN [17], GraphSAGE [36], and GAT [31], entwine feature transformation and propagation within each layer. In contrast, recent models like APPNP [18] represent Decoupled GNNs [30, 37, 38] that separate transformation and propagation. While Graph Neural Networks (GNNs) have achieved notable success across a range of domains [1], they often harbor various biases tied to node features and graph topology [39]. For example, GNNs may generate predictions skewed by sensitive node features [8, 6], leading to potential unfairness in diverse tasks such as recommendations [40] and loan fraud detection [41]. Numerous studies have proposed different methods to address feature bias, including adversarial training [8, 42, 43], and fairness constraints [6, 44, 45]. Structural bias is another significant concern, where low-degree nodes are more likely to be falsely predicted by GNNs [9]. Recently, there are several works aimed to mitigate the degree bias issue [10, 11, 12]. Distinct from these previous studies, our work identifies a new form of bias - label position bias, which is prevalent in GNNs. To address this, we propose a novel method, LPSL, specifically designed to alleviate the label position bias.

## 6 Conclusion and Limitation

In this study, we shed light on a previously unexplored bias in GNNs, the label position bias, which suggests that nodes closer to labeled nodes typically yield superior performance. To quantify this bias, we introduce a new metric, the Label Proximity Score, which proves to be a more intrinsic measure. To combat this prevalent issue, we propose a novel optimization framework, LPSL, to learn an unbiased graph structure. Our extensive experimental evaluation shows that LPSL not only outperforms standard methods but also significantly alleviates the label position bias in GNNs. In our current work, we address the label position bias only from a structure learning perspective. Future research could incorporate feature information, which might lead to improved performance. Besides, we have primarily examined homophily graphs. It would be interesting to investigate how label position bias affects heterophily graphs. We hope this work will stimulate further research and development of methods aimed at enhancing label position fairness in GNNs.

## 7 Acknowledgement

This research is supported by the National Science Foundation (NSF) under grant numbers CNS 2246050, IIS1845081, IIS2212032, IIS2212144, IIS2153326, IIS2212145, IOS2107215, DUE 2234015, DRL 2025244 and IOS2035472, the Army Research Office (ARO) under grant number W911NF-21-1-0198, the Home Depot, Cisco Systems Inc, Amazon Faculty Award, Johnson&Johnson, JP Morgan Faculty Award and SNAP.

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

# Appendix

## A Preliminary Study

In this section, we first present a comprehensive set of experimental results, showcasing the performance disparity across various GNNs concerning three metrics related to Label Distance Bias: Degree, Shortest Path Distance, and Label Proximity Score. Subsequently, we extend our examination of label position bias to additional GNNs, including ReNode [46], GCNII [47], JKNet [23], MAGNA [48], and NodeFormer [49]. We begin with an introduction to the first study.

### A.1 The performance disparity of GNNs across three metrics

**Datasets.** We selected three representative datasets for our experiments: Cora, CiteSeer, and PubMed. For each of these datasets, we worked with three different labeling rates: 5 labels per class, 20 labels per class, and 60% labels per class. For the data splits consisting of 5 and 20 labels per class, we adopted a commonly used setting [50] that randomly selects 500 nodes for validation and 1000 labels for testing. When dealing with a labeling rate of 60%, we randomly selected 20% of nodes for validation and another 20% for testing.

**Models.** Our study also incorporates three representative models: APPNP [18], GCN [17], and Label Propagation [19]. APPNP, a decoupled GNN, directly leverages the PPR matrix for feature propagation. On the other hand, GCN, a coupled GNN, uses the original adjacency matrix for feature propagation across each layer. Label Propagation relies solely on graph structure and labeled nodes for prediction. For all the models, we select their best hyperparameters based on the search space in their original papers.

**Experimental Setup.** For both Degree and Shortest Path Distance metrics, we employ their actual values, [1, 2, 3, 4, 5, 6, 7], to segregate nodes into separate sensitive groups, considering only a handful of nodes possess a degree or shortest path Distance greater than seven. We delete the groups with only a few nodes. For the Label Proximity Score (LPS), we divided the test nodes evenly into seven sensitive groups based on their LPS, each group possessing a uniform range. It's important to note that we also filtered out outliers, particularly those with significantly larger LPS than the rest. Additionally, if a group contained only a few nodes, we merged it with adjacent groups.

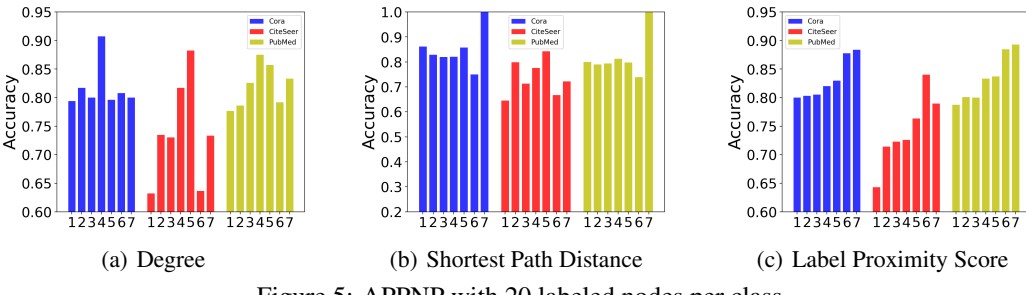

|           (a) Degree            |     (b) Shortest Path Distance     |    (c) Label Proximity Score    |

Figure 5: APPNP with 20 labeled nodes per class.

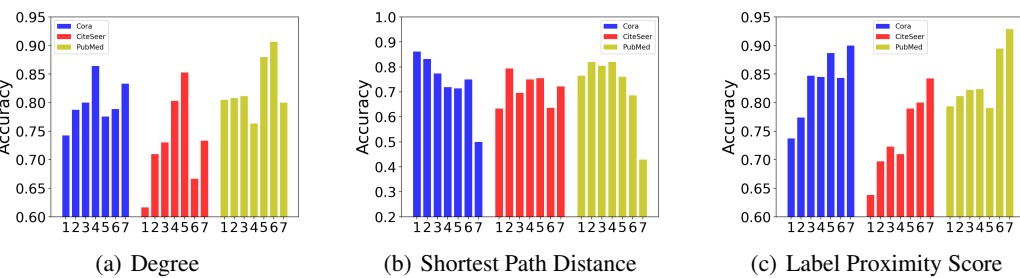

|           (a) Degree            |     (b) Shortest Path Distance     |    (c) Label Proximity Score    |

Figure 6: GCN with 20 labeled nodes per class.

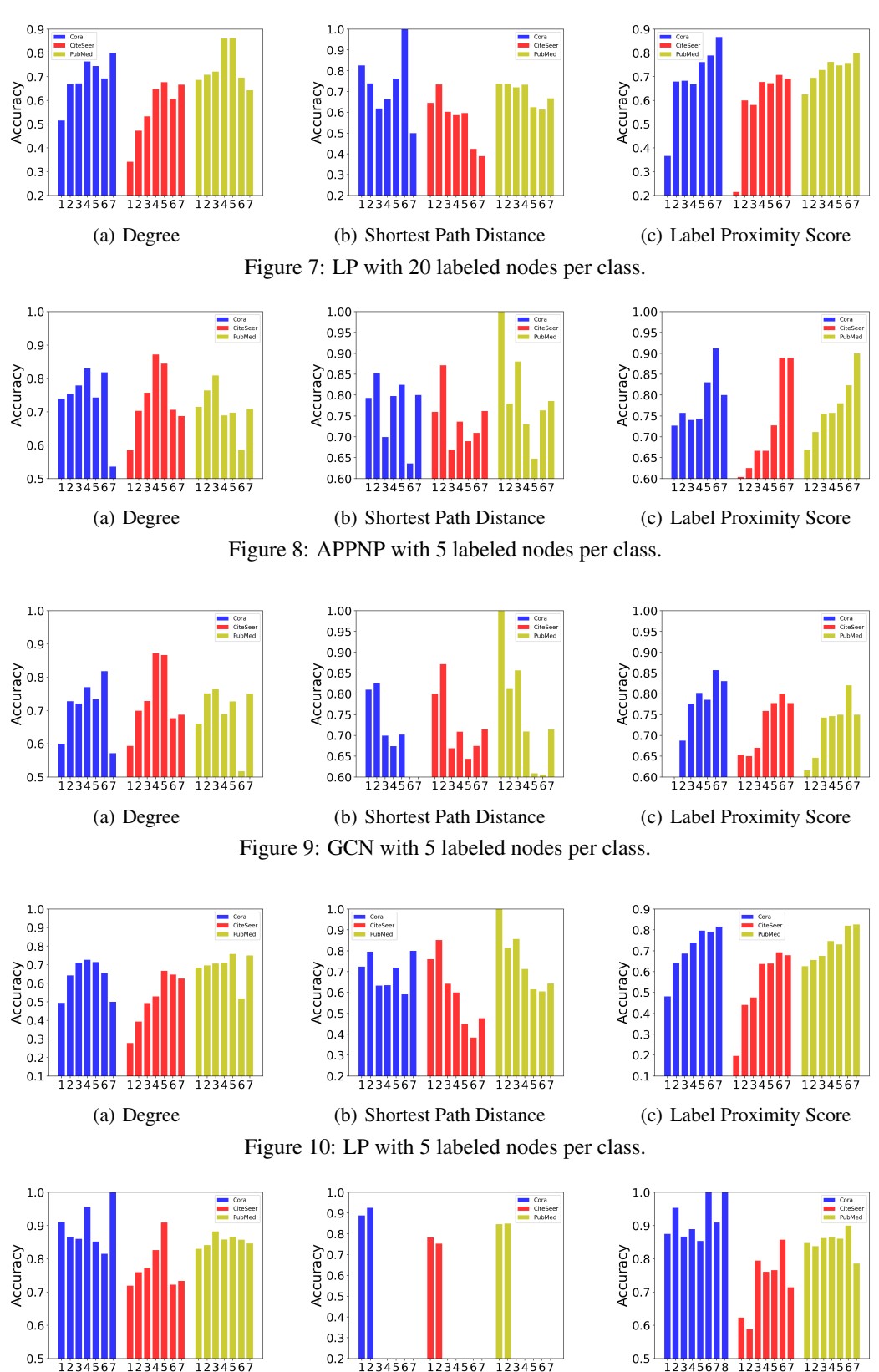

Figure 7: LP with 20 labeled nodes per class.

Figure 8: APPNP with 5 labeled nodes per class.

Figure 9: GCN with 5 labeled nodes per class.

Figure 10: LP with 5 labeled nodes per class.

Figure 11: APPNP with 60% labeled nodes per class.

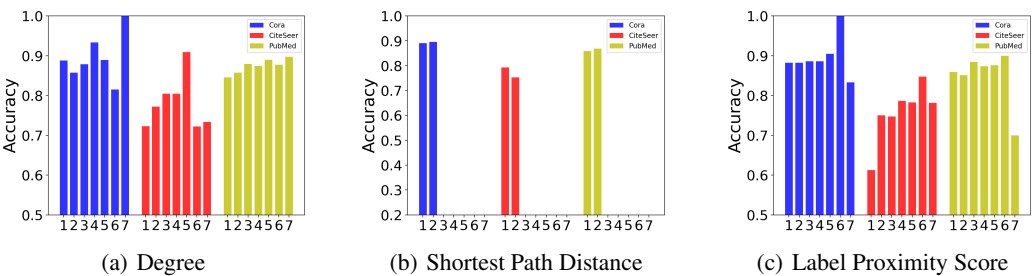

(a) Degree      (b) Shortest Path Distance      (c) Label Proximity Score

Figure 12: GCN with 60% labeled nodes per class.

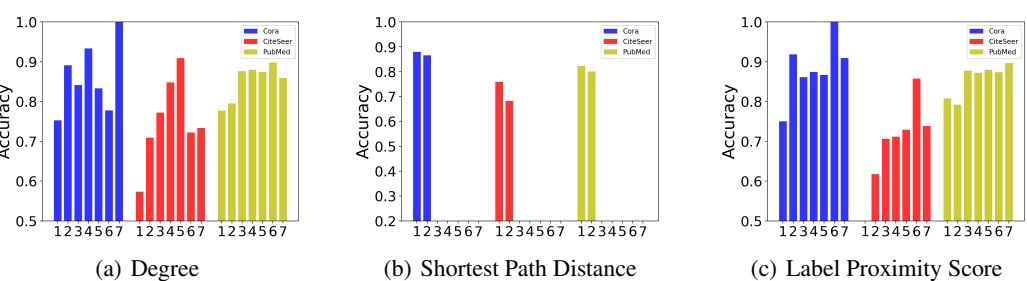

(a) Degree      (b) Shortest Path Distance      (c) Label Proximity Score

Figure 13: LP with 60% labeled nodes per class.

Our analysis of the results above leads us to the following key observations:

- Label Position Bias is a widespread phenomenon across all GNN models and datasets. Classification accuracy exhibits substantial variation between different sensitive groups, with discernible patterns.

- When contrasted with Degree and Shortest Path Distance, the proposed Label Proximity Score consistently shows a robust correlation with performance disparity across all datasets and models. This underscores its efficacy as a measure of Label Position Bias.

- The severity of Label Position Bias is more prominent when the labeling rate is low, such as with 5 or 20 labeled nodes per class. However, with a labeling rate of 60% labeled nodes per class, the bias becomes less noticeable. This is evident from the fact that the shortest path distance is either 1 or 2 for all datasets, implying that all test nodes have at least one labeled node within their two-hop neighbors.

## A.2  Label Position Bias of different GNNs

We further select 4 more GNNs, i.e., ReNode [46], GCNII [47], JKNet [23], and MAGNA [48], and 1 transformer-based model, i.e., and NodeFormer [49], to further verify the Label Position Bias.

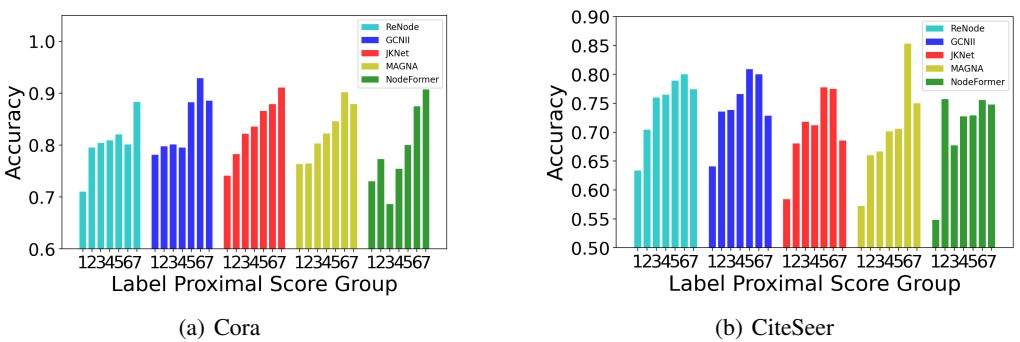

(a) Cora      (b) CiteSeer

Figure 14: The label position bias of different GNNs.

From the above results, we can find all the selected GNNs suffer from the label position bias issue. However, the graph transformer model, i.e., NodeFormer, demonstrates small label position bias on the CiteSeer dataset.

These findings highlight the importance of further exploring Label Position Bias and developing strategies to mitigate its impact on GNN performance.

# B  Understandings

*Remark 3.1* The feature aggregation using the learned graph structure $\mathbf{B}$ directly as a propagation matrix, i.e., $\mathbf{F} = \mathbf{BX}$, is equivalent to applying the message passing in GNNs using the original graph if $\mathbf{B}$ is the approximate or exact solution to the primary problem defined in Eq. (2) without constraints.

*Proof.* There are several recent studies [51, 52, 53] unified the message passing in different GNNs under an optimization framework. For instance, Ma et al. [52] demonstrated that the message-passing scheme of GNNs, such as GCN [17], GAT [31], PPNP and APPNP [18], can be viewed as optimizing the following graph signal denoising problem:

$$\operatorname*{arg\,min}_{\mathbf{F}\in\mathbb{R}^{n\times d}} \mathcal{L} = \|\mathbf{X} - \mathbf{F}\|_F^2 + \lambda\,\mathrm{tr}(\mathbf{F}^\top\tilde{\mathbf{L}}\mathbf{F}), \tag{7}$$

where $\mathbf{X} \in \mathbb{R}^{n\times d}$ is the node features, $\mathbf{F}$ is the optimal node representations after applying GNNs, and $\lambda$ are used to control the feature smoothness. The gradient of $\frac{\partial \mathcal{L}}{\partial \mathbf{F}}$ can be represented as:

$$\frac{\partial \mathcal{L}}{\partial \mathbf{F}} = 2(\mathbf{F} - \mathbf{X}) + 2\lambda\tilde{\mathbf{L}}\mathbf{F}.$$

Here, we provide two examples of using the Eq. (7) to derive APPNP and GCN. For APPNP, we can adopt multiple-step gradient descent to solve the Eq. (7):

$$\mathbf{F}^k = \mathbf{F}^{k-1} - \gamma\frac{\partial \mathcal{L}}{\partial \mathbf{F}} = (1 - 2\lambda - 2\lambda\gamma)\mathbf{F}^{k-1} + 2\lambda\gamma\tilde{\mathbf{A}}\mathbf{F}^{k-1} + 2\gamma\mathbf{X}.$$

If we set the stepsize $\gamma = \frac{1}{2(1+\lambda)}$, then we have the following update steps:

$$\mathbf{F}^k = \frac{\lambda}{1+\lambda}\tilde{\mathbf{A}}\mathbf{F}^{k-1} + \frac{1}{1+\lambda}\mathbf{X}$$

which is the message passing scheme of APPNP. Then, if we propagate $K$ layers, then

$$\begin{aligned}
\mathbf{F}^K &= \frac{\lambda}{1+\lambda}\tilde{\mathbf{A}}\mathbf{F}^{K-1} + \frac{1}{1+\lambda}\mathbf{X} \\
&= \frac{\lambda}{1+\lambda}\tilde{\mathbf{A}}\Big(\frac{\lambda}{1+\lambda}\tilde{\mathbf{A}}\mathbf{F}^{K-2} + \frac{1}{1+\lambda}\mathbf{X}\Big) + \frac{1}{1+\lambda}\mathbf{X} \\
&= \Big(\Big(\frac{\lambda}{1+\lambda}\Big)^K\tilde{\mathbf{A}}^K + \sum_{i=0}^{K-1}\frac{1}{1+\lambda}\Big(\frac{\lambda}{1+\lambda}\Big)^i\tilde{\mathbf{A}}^i\Big)\mathbf{X}.
\end{aligned} \tag{8}$$

For GCN, we can use one step gradient to solve the Eq. (7):

$$\mathbf{F} = \mathbf{X} - \gamma\frac{\partial \mathcal{L}}{\partial \mathbf{F}}\Big|_{\mathbf{F}=\mathbf{X}} = (1 - 2\gamma\lambda)\mathbf{X} + 2\gamma\lambda\tilde{\mathbf{A}}\mathbf{X}.$$

If we set the step size $\gamma = \frac{1}{2\lambda}$, then the $\mathbf{F} = \tilde{\mathbf{A}}\mathbf{X}$, which is the message passing of GCN.

The primary problem defined in Eq. (2) without constraints can be represented as:

$$\operatorname*{arg\,min}_{\mathbf{B}} \mathcal{L} = \|\mathbf{I} - \mathbf{B}\|_F^2 + \lambda\mathrm{tr}(\mathbf{B}^\top\tilde{\mathbf{L}}\mathbf{B}). \tag{9}$$

Comparing Eq. (7) with Eq. (9), the only difference lies in the first term, where the feature matrix $\mathbf{X}$ is set to be identity matrix $\mathbf{I}$. Then, we can follow the same steps to solve the Eq. (9).

If we use the multiple-step gradient descent with the stepsize $\gamma = \frac{1}{2(1+\lambda)}$, then we have the following update steps:

$$\mathbf{B}^k = \frac{\lambda}{1+\lambda}\tilde{\mathbf{A}}\mathbf{B}^{k-1} + \frac{1}{1+\lambda}\mathbf{I}.$$

Then, for $K$ steps iteration, $B^K$ will be:

$$\mathbf{B}^K = \left(\left(\frac{\lambda}{1+\lambda}\right)^K \tilde{\mathbf{A}}^K + \sum_{i=0}^{K-1}\frac{1}{1+\lambda}\left(\frac{\lambda}{1+\lambda}\right)^i \tilde{\mathbf{A}}^i\right), \tag{10}$$

which is the propagation matrix of APPNP in Eq. (8). As a result, the message passing of APPNP can be written as $\mathbf{F} = \mathbf{B}\mathbf{X}$.

If we use one-step gradient descent to solve Eq. (9), then $\mathbf{B}$ can be represented as:

$$\mathbf{B} = \mathbf{I} - \gamma\frac{\partial\mathcal{L}}{\partial\mathbf{B}}\Big|_{\mathbf{B}=\mathbf{I}} = (1 - 2\gamma\lambda)\mathbf{I} + 2\gamma\lambda\tilde{\mathbf{A}}.$$

If we set the step size $\gamma = \frac{1}{2\lambda}$, then the $\mathbf{B} = \tilde{\mathbf{A}}$. As a result, the aggregation in GCN can also be represented by $\mathbf{F} = \mathbf{B}\mathbf{X}$. $\qquad\square$

**Proposition B.1.** *The influence scores from all labeled nodes to any unlabeled node $i$ will be the equal, i.e., $\sum_{j\in\mathcal{V}_L}I_i(j) = c$, when using the unbiased graph structure $\mathbf{B}$ obtained from the optimization problem in Eq. (2) as the propagation matrix in GNNs.*

*Proof.* Following the definition in [23], the influence of node $j$ on node $i$ can be represented by $I_i(j) = sum\left[\frac{\partial\mathbf{h}_i}{\partial\mathbf{x}_j}\right]$, where $\mathbf{h}_i$ is the representation of node $i$, $\mathbf{x}_j$ is the input feature of node $j$, and $\left[\frac{\partial\mathbf{h}_i}{\partial\mathbf{x}_j}\right]$ represents the Jacobian matrix.

If we use the unbiased graph $\mathbf{B}$ as the propagation matrix, then $\mathbf{H} = \mathbf{B}\mathbf{X}$. Thus, $\mathbf{h}_{ij} = \sum_{k=0}^{n}\mathbf{B}_{ik}\mathbf{x}_{kj}$. The Jacobian matrix $\left[\frac{\partial\mathbf{h}_i}{\partial\mathbf{x}_j}\right]$ can be written as:

$$\left[\frac{\partial\mathbf{h}_i}{\partial\mathbf{x}_j}\right] = \text{Diag}([\mathbf{B}_{ij}, \mathbf{B}_{ij}, \ldots, \mathbf{B}_{ij}]), \tag{11}$$

where Diag represents the diagonal matrix. As a result, $I_i(j) = sum\left[\frac{\partial\mathbf{h}_i}{\partial\mathbf{x}_j}\right] = n\mathbf{B}_{ij}$.

Suppose the constraint $\mathbf{B}\mathbf{T}\mathbf{1}_n = \frac{c}{n}$ is satisfied, then the influence scores from all labeled nodes to the unlabeled node $i$ can be represented as:

$$\sum_{j\in\mathcal{V}_L}I_i(j) = \sum_{j\in\mathcal{V}_L}n\mathbf{B}_{ij} = n\mathbf{B}\mathbf{T}\mathbf{1}_n = c. \tag{12}$$

Finally, the influence scores from all labeled nodes to any unlabeled node $i$ are equal.

$\qquad\square$

## C  Datasets Statistics

In the experiments, the data statistics used in Section 4 are summarized in Table 6. For Cora, CiteSeer and PubMed dataset, we adopt different label rates, i.e., 5, 10, 20, and 60% labeled nodes per class, to get a more comprehensive comparison. For label rates 5, 10, and 20, we use 500 nodes for validation and 1000 nodes for testing. For label rates of 60% labeled node per class, we use half of the rest nodes for validation and the remaining half for the test. For each labeling rate and dataset, we adopt 10 random splits for each dataset. For the ogbn-arxiv dataset, we follow the original data split.

Table 6: Dataset Statistics.

| Dataset | Nodes | Edges | Features | Classes |
|---|---|---|---|---|
| Cora | 2,708 | 5,278 | 1,433 | 7 |
| CiteSeer | 3,327 | 4,552 | 3,703 | 6 |
| PubMed | 19,717 | 44,324 | 500 | 3 |
| Coauthor CS | 18,333 | 81,894 | 6,805 | 15 |
| Coauthor Physics | 34,493 | 247,962 | 8,415 | 5 |
| Amazon Computer | 13,381 | 245,778 | 767 | 10 |
| Amazon Photo | 7,487 | 119,043 | 745 | 8 |
| Ogbn-Arxiv | 169,343 | 1,166,243 | 128 | 40 |

## D  Hyperparamters Setting

In this section, we describe in detail the search space for parameters of different experiments.

For all deep models, we use 3 transformation layers with 256 hidden units for the ogbn-arxiv dataset and 2 transformation layers with 64 hidden units for other datasets. For all methods, the following common hyperparameters are tuned based on the loss and validation accuracy from the following search space:

- Learning Rate: {0.01, 0.05}
- Dropout Rate: {0, 0.5, 0.8}
- Weight Decay: {0, 5e-5, 5e-4}

For APPNP and Label Propagation, we tune the teleport probability $\alpha$ in {0.1, 0.2, 0.3, 0.4, 0.5, 0.6, 0.7, 0.8, 0.9}. For GRADE[2], we set the hidden dimension 256, the temperature in {0.2, 0.5, 0.8, 1, 1.1, 1.5, 1.7, 2}, which covers all the best values in their original paper. For SRGNN[3], we set the weight of CMD loss in {0.1, 0.5, 1, 1.5, 2}.

For the proposed LPSL, we set the $c$ in the range [0.7, 1.3], $\rho$ in {0.01, 0.001}, $\gamma$ in {0.01, 0.001}, $\beta$ in {1e-4, 1e-5, 5e-5, 1e-6, 5e-6, 1e-7}. For the LPSL$_{APPNP}$, we set $\lambda$ in {8, 9, 10, 11, 12, 13, 14, 15}. For the LPSL$_{GCN}$, we set $\lambda$ in {1, 2, 3, 4, 5, 6, 7, 8}.

## E  Node Classification Results

For the semi-supervised node classification task, we choose nine common used datasets including three citation datasets, i.e., Cora, Citeseer and Pubmed [27], two coauthors datasets, i.e., CS and Physics, two Amazon datasets, i.e., Computers and Photo [28], and one OGB datasets, i.e., ogbn-arxiv [29].

To the best of our knowledge, there are no previous works that aim to address the label position bias. In this work, we select three GNNs, namely, GCN [17], GAT [31], and APPNP [18], two Non-GNNs, MLP and Label Propagation [19], as baselines. Furthermore, we also include GRADE [32], a method designed to mitigate degree bias. Notably, SRGNN [33] demonstrates that if labeled nodes are gathered locally, it could lead to an issue of feature distribution shift. SRGNN aims to mitigate the feature distribution shift issue and is also included as a baseline. The overall performance are shown in Table 7.

## F  Ablation Study

In this subsection, we explore the impact of different hyperparameters, specifically the smoothing term $\lambda$ and the constraint $c$, on our model. We conducted experiments on the Cora and CiteSeer datasets using ten random data splits with 20 labels per class. The accuracy of different $\lambda$ and $c$ values for LPSL$_{APPNP}$ and LPSL$_{GCN}$ on the Cora and CiteSeer datasets are illustrated in Figure F

---

[2]https://github.com/BUPT-GAMMA/Uncovering-the-Structural-Fairness-in-Graph-Contrastive-Learning/
[3]https://github.com/GentleZhu/Shift-Robust-GNNs

Table 7: The overall results of the node classification task.

| Dataset | Label Rate | MLP | LP | GCN | APPNP | GAT | GRADE | SRGNN | LPSL$_{GCN}$ | LPSL$_{APPNP}$ |
|---|---|---|---|---|---|---|---|---|---|---|
| Cora | 5 | 42.34 ± 3.31 | 57.60 ± 5.71 | 70.68 ± 2.17 | 75.86 ± 2.34 | 72.97 ± 2.23 | 69.51 ± 6.79 | 70.77 ± 1.82 | 76.58 ± 2.37 | **77.24 ± 2.18** |
|  | 10 | 51.34 ± 3.37 | 63.76 ± 3.60 | 76.50 ± 1.42 | 80.29 ± 1.00 | 78.03 ± 1.17 | 74.95 ± 2.46 | 75.42 ± 1.57 | 80.39 ± 1.17 | **81.59 ± 0.98** |
|  | 20 | 59.23 ± 2.52 | 67.87 ± 1.43 | 79.41 ± 1.30 | 82.34 ± 0.67 | 81.39 ± 1.41 | 77.41 ± 1.49 | 78.42 ± 1.75 | 82.74 ± 1.01 | **83.24 ± 0.75** |
|  | 60% | 76.49 ± 1.13 | 86.05 ± 1.35 | 88.60 ± 1.19 | 88.49 ± 1.28 | 88.68 ± 1.13 | 86.84 ± 0.99 | 87.17 ± 0.95 | **88.75 ± 1.21** | 88.62 ± 1.69 |
| CiteSeer | 5 | 41.05 ± 2.84 | 39.06 ± 3.53 | 61.27 ± 3.85 | 63.92 ± 3.39 | 62.60 ± 3.34 | 63.03 ± 3.61 | 64.84 ± 3.41 | 65.65 ± 2.47 | **65.70 ± 2.18** |
|  | 10 | 47.99 ± 2.71 | 42.29 ± 3.26 | 66.28 ± 2.14 | 67.57 ± 2.05 | 66.81 ± 2.10 | 64.20 ± 3.23 | 67.83 ± 2.19 | 67.73 ± 2.57 | **68.76 ± 1.77** |
|  | 20 | 56.96 ± 1.80 | 46.15 ± 2.31 | 69.60 ± 1.67 | 70.85 ± 1.45 | 69.66 ± 1.47 | 67.50 ± 1.76 | 69.13 ± 1.99 | 70.73 ± 1.32 | **71.25 ± 1.14** |
|  | 60% | 73.15 ± 1.36 | 69.39 ± 2.01 | 76.88 ± 1.78 | 77.42 ± 1.47 | 76.70 ± 1.81 | 74.00 ± 1.87 | 74.57 ± 1.57 | 77.18 ± 1.64 | **77.56 ± 1.44** |
| PubMed | 5 | 58.48 ± 4.06 | 65.52 ± 6.42 | 69.76 ± 6.46 | 72.68 ± 5.68 | 70.42 ± 5.36 | 66.90 ± 6.49 | 69.38 ± 6.48 | 73.46 ± 4.64 | **73.57 ± 5.30** |
|  | 10 | 65.36 ± 2.08 | 68.39 ± 4.88 | 72.79 ± 3.58 | 75.53 ± 3.85 | 73.35 ± 3.83 | 73.31 ± 3.75 | 72.69 ± 3.49 | 75.67 ± 4.42 | **76.18 ± 4.05** |
|  | 20 | 69.07 ± 2.10 | 71.88 ± 1.72 | 77.43 ± 2.66 | 78.93 ± 2.11 | 77.43 ± 2.66 | 75.12 ± 2.37 | 77.09 ± 1.68 | 78.75 ± 2.45 | **79.26 ± 2.32** |
|  | 60% | 86.14 ± 0.64 | 83.38 ± 0.64 | **88.48 ± 0.46** | 87.56 ± 0.52 | 86.52 ± 0.56 | 86.90 ± 0.46 | 88.32 ± 0.55 | 87.75 ± 0.57 | 87.96 ± 0.57 |
| CS | 20 | 88.12 ± 0.78 | 77.45 ± 1.80 | 91.73 ± 0.49 | 92.38 ± 0.38 | 90.96 ± 0.46 | 89.43 ± 0.67 | 89.43 ± 0.67 | 91.94 ± 0.54 | **92.44 ± 0.36** |
| Physics | 20 | 88.30 ± 1.59 | 86.70 ± 1.03 | 93.29 ± 0.80 | 93.49 ± 0.67 | 92.81 ± 1.03 | 91.44 ± 1.41 | 93.16 ± 0.64 | 93.56 ± 0.51 | **93.65 ± 0.50** |
| Computers | 20 | 60.66 ± 2.98 | 72.44 ± 2.87 | 79.17 ± 1.92 | 79.07 ± 2.34 | 78.38 ± 2.27 | 79.01 ± 2.36 | 78.54 ± 2.15 | **80.05 ± 2.92** | 79.58 ± 2.31 |
| Photo | 20 | 75.33 ± 1.91 | 81.58 ± 4.69 | 89.94 ± 1.22 | 90.87 ± 1.14 | 89.24 ± 1.42 | 90.17 ± 0.93 | 89.36 ± 1.02 | 90.85 ± 1.16 | **90.93 ± 1.40** |
| ogbn-arxiv | 54% | 61.17 ± 0.20 | 74.08 ± 0.00 | 71.91 ± 0.15 | 71.61 ± 0.30 | OOM | OOM | 68.01 ± 0.35 | **72.04 ± 0.12** | 69.20 ± 0.26 |

and Figure F, respectively. From the results, we can find both LPSL$_{APPNP}$ and LPSL$_{GCN}$ are not very sensitive to $\lambda$ and $c$ at the chosen regions.

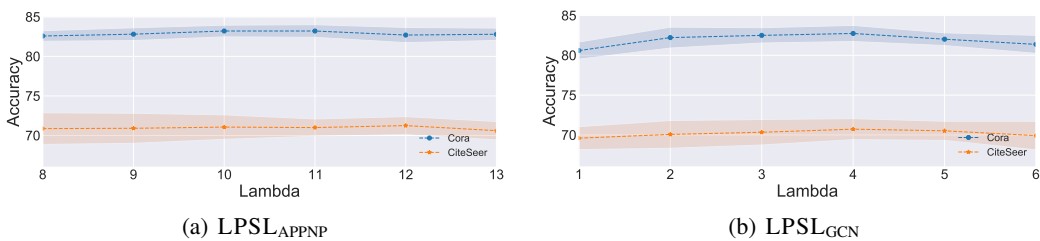

(a) LPSL$_{APPNP}$      (b) LPSL$_{GCN}$

Figure 15: The accuracy of different $\lambda$ for LPSL$_{APPNP}$ and LPSL$_{GCN}$ on Cora and CiteSeer datasets.

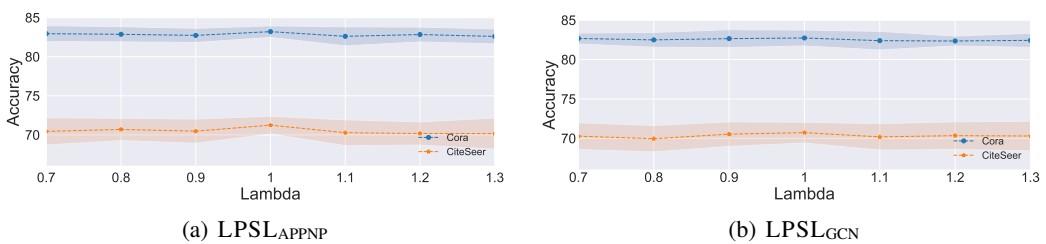

(a) LPSL$_{APPNP}$      (b) LPSL$_{GCN}$

Figure 16: The accuracy of different $c$ for LPSL$_{APPNP}$ and LPSL$_{GCN}$ on Cora and CiteSeer datasets.

