# OpenReview forum: "Towards Label Position Bias in Graph Neural Networks"
_NeurIPS.cc/2023/Conference — NeurIPS 2023 poster_

### Official Review · Reviewer_tP9u · 2023-06-25

**Soundness:** 3 good
**Presentation:** 3 good
**Contribution:** 3 good
**Rating:** 5
**Confidence:** 3

**Summary:**

This paper presents a study on Label Distance Bias in Graph Neural Networks (GNNs). The authors investigate the performance disparity across various GNN models on three datasets and analyze three metrics related to Label Distance Bias. The study reveals that Label Position Bias is a significant factor affecting the performance of GNNs. The proposed Label Proximity Score consistently exhibits a robust correlation with performance disparity. The findings highlight the importance of further exploring Label Position Bias and developing strategies to mitigate its impact on GNN performance. The paper provides valuable insights into the performance variation of GNNs and underscores the need for addressing Label Distance Bias to enhance the effectiveness of GNN models.

**Strengths:**

This paper offers three key advantages in its study on Label Distance Bias in Graph Neural Networks (GNNs).

Firstly, it identifies Label Position Bias as a widespread phenomenon across all GNN models and datasets, highlighting the importance of understanding and addressing this bias in GNN performance.

Secondly, the paper introduces the Label Proximity Score as a metric that consistently shows a robust correlation with performance disparity. This score proves to be an effective measure of Label Position Bias, outperforming Degree and Shortest Path Distance metrics.

Lastly, the study demonstrates that the severity of Label Position Bias is more prominent at lower labeling rates, but becomes less noticeable when 60% of nodes per class are labeled. This finding provides insights into the impact of labeling rates on bias and suggests strategies for mitigating its effects.

**Weaknesses:**


Although your paper presents well and the proposed method seems to be very targeted towards relevant issues, I cannot determine the actual performance of your work as you have not provided any code.

**Questions:**

 1. Can the proposed method effectively mitigate label position bias in datasets with imbalanced class distributions?

2.  What are the limitations of the selected baseline models (GCN, GAT, APPNP, MLP, Label Propagation) in addressing label position bias?

3.  How does the proposed method compare to existing methods in terms of computational efficiency?

 4. Can the proposed method be applied to larger and more complex datasets, and if so, how does its performance scale?






**Limitations:**

yes

---

> ### Author Rebuttal · Authors · 2023-08-10
>
> Dear reviewer,
>
> We appreciate your constructive feedback. We are pleased to provide detailed responses to address your concerns.
>
> **Q1: I cannot determine the actual performance of your work as you have not provided any code.**
>
> **A1:** Thank you for highlighting your concern. We apologize for any confusion. While the code was not directly included in the Supplementary Material, we have provided an anonymous link to the code in Appendix D of our paper.
>
> **Q2: Can the proposed method effectively mitigate label position bias in datasets with imbalanced class distributions?**
>
> **A2:** Thanks for your thoughtful consideration. Our method is primarily designed to address biases stemming from the positions of labeled nodes rather than the categories to which these nodes belong. As such, the class distribution, whether balanced or imbalanced, should not directly impact our method's efficacy in mitigating label position bias.
>
> To empirically validate this claim, we conducted tests on datasets with imbalanced class distributions, specifically Cora and CiteSeer. Adhering to the setup described in [1], we designated half of the classes as major classes, endowing them with five times as many labeled nodes as the minor classes. The outcomes of these tests are detailed in Table 3 of our rebuttal document. The results consistently show that our method not only enhances overall performance but also effectively curtails label position bias, even in imbalanced scenarios.
>
>
> [1] Chen, Deli, et al. ”Topology-imbalance learning for semi-supervised node classification, NeurIPS 2021.
>
> **Q3: What are the limitations of the selected baseline models in addressing label position bias?**
>
> **A3:** Our investigation has revealed that label position bias is prevalent across a variety of GNNs. (More results can be found in Figure 1 of the rebuttal document) Notably, prior GNN models, including but not limited to GCN, GAT, APPNP, and Label Propagation, exhibit this bias. These models, in their original formulations, do not provide mechanisms to specifically address or mitigate label position bias.
>
>
> **Q4: How does the proposed method compare to existing methods in terms of computational efficiency?**
>
> **A4*:* The essence of our proposed method is to produce a fair graph structure that is seamlessly integrable with existing GNNs. Our method won't change the inherent computational processes of the backbone models. However, if the B is a dense matrix, then it would affect the computational efficiency. As a result, we design a $l_1$-regularized Label Position Unbiased Sparse Structure Learning method to generate a sparse matrix. With the sparse matrix B, there's no significant impact on the backbone models' computational efficiency.
>
>
> **Q5: Can the proposed method be applied to larger and more complex datasets, and if so, how does its performance scale?**
>
> **A5*:* Yes, our proposed method can be applied to larger datasets. For instance, it has been tested and found applicable on the Ogbn-Arxiv dataset. However, it's worth noting that the formulation in Eq. 2 of our paper tends to produce a dense matrix, which might not be optimal for large-scale graphs. To address this challenge, we've introduced the $l_1$-regularized Label Position Unbiased Sparse Structure Learning. To further bolster efficiency, especially for large graphs, we employ the Block Coordinate Descent (BCD) method. This approach is particularly advantageous as it allows us to update just a segment of the structure matrix at any given time, enhancing scalability for larger datasets.
>
> We hope that we have addressed the concerns in your comments, and please
> kindly let us know if there is any further concern, and we are happy to clarify.

---

> > ### Author Response · Authors · 2023-08-18
> > **A Gentle Remind to Reviewer tP9u**
> >
> > Dear Reviewer tP9u,
> >
> > We would like to express our sincere gratitude to you for reviewing our paper and providing valuable feedback. Could we kindly know if the responses have addressed your concerns? If there are any further questions, we are happy to clarify. Thank you.
> >
> > Best,
> >
> > All authors

---

### Official Review · Reviewer_fFnJ · 2023-06-28

**Soundness:** 3 good
**Presentation:** 3 good
**Contribution:** 3 good
**Rating:** 6
**Confidence:** 5

**Summary:**

This paper demonstrates a new bias in GNN models called label position bias. A metric called the Label Proximity Score is introduced to quantify the bias. The paper proposes a framework to learn an unbiased graph structure, which can be applied to existing GNN models.

**Strengths:**

Label position bias is an interesting phenomenon uncovered by the authors. This has important fairness implications in GNN model training.

**Weaknesses:**

1. The label position bias is demonstrated empirically. It would be good if the authors can give an explanation for why this phenomenon occurs. Is this issue present in *all* GNN models inherently?

1. There are no statistical significance tests done to verify that the observed label position bias is indeed statistically significant.

1. More insights can be gained if the LPSL unbiased B can be tested using more diverse GNN models.

**Questions:**

1. In the LPSL problem in (2), is there any constraint that the entries of B have to be non-negative? How do you interpret negative entries?

2. The constraint in (2) mimics (1) but the optimal solution B is not equal to P (unless the constraint is absent). How does B mitigate the LPS issue identified in Figure 3? There are no experimental results (taking the optimal B found as the adjacency matrix) showing this, which is the motivation for the paper.

3. The definition of the influence I_i(j) is incorrect. From [23], it should be the sum of the absolute values of the entries of the Jacobian. How does this affect the proof in the appendix? Furthermore, it is difficult for the reader to guess what "sum" notation means.

4. Same issue in (5) as Q1.

5. Is label position bias apparent in heterophily graph datasets like Chameleon and Squirrel?



**Limitations:**

N.A.

---

> ### Author Rebuttal · Authors · 2023-08-10
>
> Dear reviewer,
>
> We appreciate your constructive feedback. We are pleased to provide detailed responses to address your concerns.
>
> **Q1: Why the label position bias phenomenon occurs?**
>
> **A1:** Thanks for your thorough consideration. We attribute the root cause of label position bias to the inherent message passing in most GNNs. Please refer to the **Common Concern 1** in our global response.
>
> **Q2: Whether the observed label position bias is indeed statistically significant?**
>
> **A2:** To validate the statistical significance of the observed label position bias, we undertook rigorous testing. Experiments were conducted using GCN and APPNP models on both the Cora and CiteSeer datasets, with each experiment being executed
> 10 times. Subsequently, a t-test was performed for each pair of groups. Figure 2 of our rebuttal document shows that the
> majority of p-values fall below the 0.05 threshold. Only a few of adjacent groups deviated from this trend. These results strongly affirm that the label position bias is indeed statistically significant.
>
> **Q3: More insights can be gained if the LPSL unbiased B can be tested using more diverse GNN models.**
>
> **A3:** Thanks for the suggestions. We add the GCNII and JKNet backbones. Please refer to the **Commen Concern 2** in our global response.
>
>
> **Q4: Does the matrix B in Eq. (2) have negative entries?**
>
> **A4:** Thank you for your thoughtful question. The matrix B should be non-negative as it represents a fair graph structure. And the non-negative constraint should be added to our problem. However, in our practical observations, most B matrices do not contain negative entries. For instance, within our explored hyperparameter space for the Cora dataset, no instances of B had negative values. In the case of the CiteSeer dataset, negative entries in B were exceedingly rare, occurring only when the coefficient c in Eq.(2) was particularly small. Even then, the largest negative value observed was a mere -0.002. For a clearer perspective, we present the following results comparing scenarios with and without negative values:
>
> |        | With Negative | Without Negative |
> |--------|---------------|------------------|
> | Case 1 | 70.82 ± 1.50  | 70.79 ± 1.55     |
> | Case 2 | 70.45 ± 1.48  | 70.60 ± 1.57     |
>
> The performance gap between scenarios with and without negative values is minimal.
>
> **Q5: How does B mitigate the LPS issue?**
>
> **A5:** The learned fair graph structure B can ensure each test node possesses an identical LPS. By utilizing the optimal B during the propagation process, we ensure that each node receives consistent influence from all labeled nodes, thereby addressing the LPS issue.
>
> To verify that the learned B can mitigate the LPS issue, we adopt three representative group bias measurements, i.e., WDP, WSD, and WCV. The small value means that the performance difference between groups is small, which is more fair. From the results in Table 2 of our main paper and Table 1 of the rebuttal pdf, our proposed method can alleviate the LPS for all GNNs, including GCN, APPNP, GCNII, and JKNet.
>
> **Q6: How does the lack of absolute operator for the Jacobian affect the proof in the appendix? The "sum" notation is confusing.**
>
> **A6:} We appreciate your observation regarding the definition of the influence $I_i(j)$. However, as stated in $A4$, the B should be non-negative. With a positive B matrix, the omission of the absolute operator does not impact the proofs provided.
>
> We will ensure clarity in our revision to avoid any confusion regarding the "sum" notation or other aspects.
>
> **Q7: Is label position bias apparent in heterophily graph datasets like Chameleon and Squirrel?**
>
> **A7:** We undertook tests to assess the label position bias on both the Chameleon and Squirrel datasets. The findings, as detailed in Figure 3 of our rebuttal document, are quite interesting.
>
> For the Chameleon dataset, nodes exhibiting higher LPS paradoxically tend to underperform, aligning with the heterophily hypothesis. Conversely, nodes with a lower LPS also demonstrate diminished performance. This suggests that the label position bias is indeed conspicuous within the Chameleon dataset. However, such a bias is not obvious in the Squirrel dataset as all groups have similar performances. We will do further explorations for the LPS issue on the heterophily datasets.
>
> We hope that we have addressed the concerns in your comments, and please
> kindly let us know if there is any further concern, and we are happy to clarify.

---

> > ### Comment · Reviewer_fFnJ · 2023-08-11
> >
> > Thank you for your clarifications. To ensure correctness, I will require that the non-negativity constraint is added to the problem formulation (2) and the optimization steps (3)-(4) be updated to incorporate this constraint.

---

> > > ### Author Response · Authors · 2023-08-11
> > >
> > > Dear Reviewer fFnJ,
> > >
> > > We sincerely appreciate your timely feedback.
> > >
> > > In light of your suggestions, we will incorporate the non-negativity constraint in all related formulations during our revision. Besides, for the experiments results, we've verified that the matrix B under optimal hyperparameters does not contain any negative entries.
> > >
> > > Furthermore, we recognize that our response in **A4** might have lacked clarity. To elucidate, in Answer 4, 'Case 1' and 'Case 2' refer to two distinct sets of hyperparameters. The terms 'With/Without Negative' are indicative of the presence or absence of the non-negativity constraint under identical hyperparameters.
> > >
> > > Thank you once again for your invaluable insights. If you have any further questions, please kindly let us know.
> > >
> > > Best,
> > >
> > > All authors

---

> > > > ### Author Response · Authors · 2023-08-13
> > > > **Further Response about the Non-negativity Constraint**
> > > >
> > > > Dear Review fFnJ,
> > > >
> > > > We would like to provide further details  and discussions about the non-negativity constraint.  In this response, we will first elaborate on how we integrated this constraint into our model. Subsequently, we will present findings to demonstrate that the absence of this constraint has not influenced our experimental results.
> > > >
> > > > With the incorporation of the non-negativity constraint, the optimization problem is described as:
> > > > $$
> > > > \begin{aligned}
> > > > & \underset{\mathbf{B}}{\arg \min }\|\mathbf{I}-\mathbf{B}\|_F^2+\lambda \operatorname{tr}\left(\mathbf{B}^{\top} \tilde{\mathbf{L}} \mathbf{B}\right) \\\\
> > > > & \text { s.t. } \quad \mathbf{B} \mathbf{T} \mathbf{1}_n=c \mathbf{1}_n, \quad\mathbf{B}\ge0
> > > > \end{aligned}
> > > > $$
> > > >
> > > > To tackle this problem, we employed the Lagrange Multiplier method coupled with projected gradient descent. Specifically, the iterations for $\mathbf{B}$ are:
> > > > $$
> > > > \begin{aligned}
> > > > & \mathbf{B}^{k+1}=\underset{\mathbf{B}}{\arg \min } L_\rho\left(\mathbf{B}^k, \mathbf{y}^k\right) \\\\
> > > > & \mathbf{B}^{k+1} = max(0, \mathbf{B}^{k+1}) \\\\
> > > > & \mathbf{y}^{k+1}=\mathbf{y}^k+\rho\left(\mathbf{B}^{k+1} \mathbf{T} \mathbf{1}_n-c \mathbf{1}_n\right),
> > > > \end{aligned}
> > > > $$
> > > >
> > > > where $L_\rho(\mathbf{B}, \mathbf{y})=\|\mathbf{I}-\mathbf{B}\|_F^2+\lambda \operatorname{tr}\left(\mathbf{B}^{\top} \tilde{\mathbf{L}} \mathbf{B}\right)+\mathbf{y}^{\top}\left(\mathbf{B} \mathbf{T} \mathbf{1}_n-c \mathbf{1}_n\right)+\frac{\rho}{2}\left\|\mathbf{B} \mathbf{T} \mathbf{1}_n-c \mathbf{1}_n\right\|_2^2$ is the augmented Lagrange function.
> > > >
> > > > From previous discussions, we've made several observations:
> > > >
> > > > 1. Only when the hyperparameter **c** is extremely small does $\mathbf{B}$ contain a few minor negative entries.
> > > > 2. As evidenced in **A4**, introducing the non-negativity constraint has a negligible impact on performance under the same hyperparameter settings.
> > > > 3. In our experiments, the optimal $\mathbf{B}$ was found to be non-negative under the best hyperparameters.
> > > >
> > > > We further test whether the non-negative $\textbf{B}$ would be changed after applying the non-negativity constrain. Here is the finding:
> > > >
> > > > 4. For non-negative $\textbf{B}$ derived without the non-negativity constraint, reintroducing the constraint didn't alter $\textbf{B}$, ensuring consistent performance.
> > > >
> > > > These findings indicate that the absence of the non-negativity constraint has not substantially influenced our experimental results.
> > > >
> > > > Of course, the non-negativity constrain is essential, and we will add it into our revision. Thanks again.
> > > >
> > > > We hope this provides clarity on the matter. If you have any further questions, we are happy to discuss.
> > > >
> > > >
> > > >
> > > > Best,
> > > >
> > > >
> > > >
> > > > All authors.

---

### Official Review · Reviewer_Lj4a · 2023-07-01

**Soundness:** 3 good
**Presentation:** 3 good
**Contribution:** 2 fair
**Rating:** 5
**Confidence:** 5

**Summary:**

This paper pointed out that message-pass based GNNs training usually suffers from label position bias problem for semi-supervised node classification especially for quite few-shot settings. That is, the unlabeled nodes which far away from the labeled ones might suffer from performance degradation.

The paper first proposed a new metric named “Label Proximity Score” based on PPR matrix to measure label position bias, and then put forward a label unbiased graph learning method by learning a dense graph structure to guide the graph representation learning, the dense matrix will mitigate the label position bias.

**Strengths:**

1.Induced bias in GNNs for graph representation learning is one challenge problem which affects the fairness or robustness of down-streaming supervised learning tasks. The paper first found out the phenomenon that label position bias in GNNs (especially for one-hop neighbor message passing based GNNs) for node classification task. A quantified metrics is proposed to measure label position bias.

2.A position unbiased graph representation learning framework is proposed by learning a dense matrix to guide feature aggregation to mitigate the position bias. Meanwhile, the paper also gave analysis of the induced dense matrix to connect with existing PPR based APPNP.

**Weaknesses:**

1.Lack of the analysis of the root to cause label position bias in GNNs over node classification. For example, the nodes being far away from the labeled ones requires deep GNNs to receive label information. The performance degradation might due to shallow GNNs with limited receipt field or deep GNNs suffering from over-smoothing or over-squashing.

2.More backbone GNNs should be evaluated to support the phenomena of Label Position Bias. GCN is a kind of one-hop neighboring based message passing which might suffer from over-smoothing to multi-hop neighbor information aggregation. Although APPNP can aggregate multi-hop neighbor information, it is sensitive to selection of iteration number K and re-start probability parameter alpha.

It is better to select some GNNs (such as JKNet, Multi-hop Attention Neural Networks etc.) which is robust to over-smoothing or other dense matrix based GNNs (such as Graph Transformer)

**Questions:**

1.Introducing l1 norm will sparse the induced matrix B and increase the computation efficiency, however, it seems that sparsity also diffs different nodes. Does sparsity make a tradeoff to relieve label position bias? Does this sparsity find better aggregation patterns? Does unbiased graph structure mean a dense induced matrix?
2. How to get the sensitive groups with respect to different bias metrics (Degree, Shortest Path Distance and Label Proximity scores)? Is the distribution of labeled node number balance among different groups?
3.The performance of GNNs over graph with few node labels heavily depends on the graph split, how to split the graphs to conduct experiments in this paper?

**Limitations:**

1.Backbone GNNs are limited and lack of deep insight of the cause of label position bias.
2.Some important references are missing. For example, it is noted that GNNs might be not sensitive to position representation learning (Position-aware Graph Neural Networks). This is another kind of position bias, what is the connection of label position bias to such position embedding learning?

---

> ### Author Rebuttal · Authors · 2023-08-10
>
> Dear reviewer,
>
> We appreciate your constructive feedback. We are pleased to provide detailed responses to address your concerns.
>
> **Q1: What is the root to cause the label position bias in GNNs? The performance degradation might due to shallow GNNs or deep GNNs suffering from over-smoothing or over-squashing.**
>
> **A1:** Thanks for your thorough consideration. We attribute the root cause of label position bias to the inherent message passing in most GNNs. Please refer to the **Common Concern 1** in our global response.
>
> Besides, the performance degradation may not be due to the shallow GNNs or the deep GNNs with over-smoothing. First, both shallow and deep GNNs suffer from label position bias. Second, deep GCNII and JKNet are not suffering from over-smoothing, but they still have bias issues. Whether the reason is due to the over-squashing is still under exploration. However, the message passing with the unfair graph structure should be one of the reasons.
>
> **Q2: More backbone GNNs should be evaluated to support the phenomena of Label Position Bias.**
>
> **A2:** We've selected 4 more GNNs, i.e., ReNode, GCNII, JKNet, and MAGNA (Multi-hop Attention Neural Networks),  and 1 transformer-based model, i.e., NodeFomer. From Figure 1 in our rebuttal pdf file, all GNNs suffer from the label position bias issue. However, the NodeFomer has a small label position bias. More explanation can be found in the **Common Concern 1**.
>
> **Q3: Does sparsity make a tradeoff to relieve label position bias?**
>
> **A3:** Thanks for this great suggestion that motivates us to explore the relationship between sparsity and label position bias by conducting experiments using both APPNP and GCN models on the Cora datasets, varying the sparsity of matrix B.
> Table 2 of our rebuttal pdf file shows that fairness metrics remained consistent across all sparsity levels. This suggests that the sparsity introduced by the l1 norm doesn’t significantly impact our method’s efficacy in addressing
> label position bias.
>
> **Q4: Does this sparsity find better aggregation patterns?Does unbiased graph structure mean a dense induced matrix?**
>
> **A4:** In Table 2 of the rebuttal pdf file, we can conclude that sparsity may affect the model performance. The
> accuracy with 80\% and 90\% sparsity closely aligns with that of the dense matrix. However, as sparsity approaches 95\%,
> there’s a minor decline in accuracy, similar to the results of PPRGo [1].
>
> A sparse structure matrix can also be an unbiased graph structure. We can use the difference of the label position score among different nodes to measure whether a graph structure is fair or not. During the experiments, we find the loss $\|\mathbf{B T} \mathbf{1}_n -c \mathbf{1}_n\|^2 < 0.05$ on Cora dataset, which is comparable to the dense B matrix.
>
>
> [1] Bojchevski, Aleksandar, et al. "Scaling graph neural networks with approximate pagerank." KDD 2020.
>
> **Q5: How to get the sensitive groups? Is the distribution of labeled node numbers balanced among different groups? **
>
> **A5:** We categorize nodes into sensitive groups based on the same interval length related to the specific metrics. For both degree and shortest path distance, we employ an interval of 1 as the criterion. This means that nodes sharing the same degree or shortest path distance are grouped together, forming a distinct sensitive group. For the Label Proximal Score (LPS), which is a continuous value, we also adopt a uniform interval approach. Specifically, we determine the maximum and minimum LPS values and divide this range into equal-length intervals, placing nodes into groups accordingly.
>
> We kindly point out that labeled nodes are not categorized into sensitive groups. Our primary objective in utilizing sensitive groups is to assess the performance of test nodes. Moreover, due to our use of equal-length intervals, the distribution of unlabeled nodes across different groups is not balanced.
>
>
> **Q6: How to split the graphs to conduct experiments?**
>
> **A6:** The choice of data splits can indeed influence models' performance. As a result, we employ 10 random data splits for each dataset in our experiments. For every data split, we train the model 3 times to ensure robust and consistent results.
>
> **Q7: What is the connection of label position bias to position embedding learning?**
>
> **A7:** The position embedding learning aims to distinct nodes with the same structure, which may get the same representations by GNNs. It can improve the expressiveness of GNNs and is useful for tasks that require unique representations for node pairs, such as link prediction. The label position bias zeroes in on the unfairness introduced due to the positions of labeled nodes in the node classification task. It is mainly due to the unfair graph structure and unrelated to the expressive power of GNNs.
>
> We hope that we have addressed the concerns in your comments, and please
> kindly let us know if there is any further concern, and we are happy to clarify.

---

> > ### Author Response · Authors · 2023-08-18
> > **A Gentle Remind to Reviewer Lj4a**
> >
> > Dear Reviewer Lj4a,
> >
> > We would like to express our sincere gratitude to you for reviewing our paper and providing valuable feedback. Could we kindly know if the responses have addressed your concerns? If there are any further questions, we are happy to clarify. Thank you.
> >
> > Best,
> >
> > All authors

---

> > ### Comment · Reviewer_Lj4a · 2023-08-18
> >
> > Thanks for the detailed clarification from authors. I have carefully read the rebuttal. The response answers most of my concerns, and thanks for additional experiments on more backbone GNNs to verify the universality of label position bias. I decided to raise my score.

---

> > > ### Author Response · Authors · 2023-08-18
> > > **Thanks for your response**
> > >
> > > Dear Reviewer Lj4a,
> > >
> > > Thanks for your response and support. We appreciate the insights you have provided and will incorporate them into our revised manuscript.
> > >
> > > Best regards,
> > >
> > > All Authors

---

### Official Review · Reviewer_VK13 · 2023-07-06

**Soundness:** 3 good
**Presentation:** 3 good
**Contribution:** 2 fair
**Rating:** 5
**Confidence:** 4

**Summary:**

This paper presents a new bias problem on graph neural networks: label position bias, which implies that nodes perform better closer to labeled nodes. A new metric - the Label Proximity Score(LPS) is proposed to quantify this bias, and a new optimization framework is proposed to learn a new unbiased graph structure. This framework motivates all nodes to have the similar LPSs and is experimentally shown to alleviate the label position bias problem and improve model performance.


**Strengths:**

1) This paper mines a novel problem on GNNs: label position bias. This is a valuable problem, which can help us better understand the role of labeled nodes for test nodes.
2) The newly proposed metric label proximity score can measure the importance and influence of test nodes and labeled nodes, and the experiments in the paper show that certain trends can be identified by the label similarity score compared to the degree and shortest path distance.
3) The extensive experimental evaluation shows that LPSL not only outperforms standard methods but also significantly alleviates the label position bias in GNNs. This work could stimulate further research and development of methods aimed at enhancing label position fairness in GNNs.


**Weaknesses:**

1) The proposed label position bias problem is similar to Topology-Imbalance Node Representation Learning, which has not been mentioned in related works, and I do not understand the difference between the two problems.

Chen, Deli, et al. "Topology-imbalance learning for semi-supervised node classification." Advances in Neural Information Processing Systems 34 (2021): 29885-29897.

2) In the design of the framework only the new unbiased graph structure is calculated and its contribution may not be sufficient enough to even do ablation experiments to verify the role of the different parts.
3) The authors believe that the graph structure can solve the label position bias problem by ensuring that the LBS of each node is similar in the proposed framework, in order to fairness of all nodes. But there is confusion about this, is it possible that unfair LBS will get better results? Each node has different structural characteristics and has different relationships with labeled nodes. For example, for nodes with high degree and strong association with labeled nodes, or nodes with high degree and weak association with labeled nodes, is it necessary to distinguish between the former type of nodes, whose labels are more credible. Does the model get better performance if the degree or the shortest path distance is taken into account together with the label position bias problem?

Typos:
1) Reuse of symbols. The lowercase letter c is defined as the number of classes in Section 2, which in turn is defined as the hyperparameter serving as the uniform Label Proximity Score in Equation 2.
2) In line 312 the reference of the figure should be 4 instead of 4.4.


**Questions:**

Please see the above

**Limitations:**

In the paper, the authors mention that the existing methods may not scale to heterophily graphs.

---

> ### Author Rebuttal · Authors · 2023-08-10
>
> Dear reviewer,
>
> We appreciate your constructive feedback. We are pleased to provide detailed responses to address your concerns.
>
> **Q1: What is the difference between the Topology-Imbalance Node Representation Learning (TINL) and our proposed Label Position Bias (LPB) problem?**
>
> **A1:**  **TINL** operates on the premise that the labeled nodes have different importance with respect to the decision boundaries.
> Specifically, if two nearby labeled nodes belong to different classes, they might exert conflicting influences. The ReNode method will decrease the weight of these nodes. However,  **LPB** doesn’t hinge on the classes of the labeled nodes or the different importance of labeled nodes. Instead, it focus on the performance disparity of different groups of test node with different proximal distance to the labeled nodes.
>
> Although both TINL and LPB consider the influence of labeled nodes, they focus on different perspectives. TINL
> grapples with influence conflicts due to class disparities of neighboring labeled nodes. Conversely, LPB delves into biases
> triggered by varying levels of influence from the labeled nodes.
>
> We also conduct experiments using the proposed ReNode method for TINL. In Figure 1 of the rebuttal pdf file, the ReNode method still suffers from the label position bias problem.
>
> **Q2: The contribution is insufficient only with the new unbiased graph structure learning.**
>
> In this work, we first identify a novel bias in GNNs: the Label Position Bias (LPB), quantifiable using our introduced label proximal score. This bias results in unequal predictions for distinct test nodes, which could lead to an unfairness issue. To alleviate the problem, we propose a
> simple yet efficient method to learn an unbiased graph structure. However, graph structure learning would lead to a dense matrix, which limits the graph scale. We design an efficient sparse graph structure learning algorithm, which can be applied to large graphs. The learned unbiased graph structure can be easily incorporated into different GNNs. We add more GNN backbones, i.e., GCNII and JKNet. From the results in Tabel 2 of the main paper and Table 1 of the rebuttal pdf file, our method not only can improve the overall performance but also alleviate the label position bias and improve fairness over different GNNs.
>
> **Q3: Is it possible that unfair LPS will get better results?**
>
> **A3:** Yes, it is possible that an unfair label proximal score (LPS) could have a higher performance. The LPS can measure the proximal distance of a test node to all labeled nodes, which can be understood as the influence from labeled nodes. A fair LPS means the graph structure is fair with respect to the position of labeled nodes. However, a fair graph structure can't guarantee better performance. For example, a fully connected graph may not work well, although it is fair. During our experiments, we found some graph structures can lead to lower label position bias, but result in a worse performance. Here is one example:
>
> |             | ACC          | WDP    | WSD    | WCV    |
> |-------------|--------------|--------|--------|--------|
> | Original    | 82.34 ± 0.67 | 0.0497 | 0.0616 | 0.0732 |
> | FairGraph 1 | 81.90 ± 1.13 | 0.0322 | 0.0411 | 0.0503 |
> | FairGraph 2 | 83.24 ± 0.75 | 0.039  | 0.0476 | 0.0562 |
>
> The learned FairGraph1 perform worse than the original unfair graph but has a better fairness respect to the label position bias.
>
> **Q4: Does the model get better performance if the degree or the shortest path distance
> is taken into account together with the label position bias problem?**
>
> **A4:** It is highly possible that our model can perform better, incorporating some methods that can alleviate the degree or the shortest path distance. However, there is no work to address the shortest path distance bias. And the works related to degree bias all need the discrete degree. Thus, it is hard to combine the proposed LPSL with these methods.
>
> Indeed, the proposed label proximal score already takes the degree and shortest path distance into account. The node with a higher degree and shorter distance to labeled nodes tends to have a higher label proximal score. Our proposed method can make sure each node has the same label proximal score, which implicitly addresses the degree and shortest path distance bias. Experimental results in Tables 3 \&4 of the main paper demonstrate our method can mitigate the degree and shortest path distance bias.
>
> **Q5: Typos.**
>
> **A5*:* Thank you for highlighting the Typos. We appreciate the feedback and will thoroughly review and revise our paper to address them.
>
> We hope that we have addressed the concerns in your comments, and please
> kindly let us know if there is any further concern, and we are happy to clarify.

---

> > ### Comment · Reviewer_VK13 · 2023-08-15
> >
> > The reviewer appreciates the responses from the authors, which have addressed most concerns from the reviewer. Further, based on the responses, I want to add a new more question:
> >
> > Since unfair LPS could get better results, why do we need to alleviate label position bias and take it as a metric to show the advantage of GNNs?

---

> > > ### Author Response · Authors · 2023-08-16
> > >
> > > Dear Reviewer VK13:
> > >
> > > Thanks for your prompt reply and insightful question. The label position bias, where nodes distant from labeled nodes tend to underperform, is a significant concern especially when GNNs are applied in real-world scenarios. For instance, in applications like fraud detection where labeled data might be scarce, users far away from these labeled nodes are at a higher risk of being misclassified due to the label position bias, leading to potential unfairness. Therefore, while it's true that fair LPS doesn't mean better results, it's crucial to ensure fairness across the graph for the broader applicability and trustworthiness of GNNs.
> > >
> > > There is usually a trade-off between the fairness and model performance [1]. Improving fairness can sometimes come at the cost of performance. However, our experiments on multiple datasets show that our proposed method not only improves fairness but also maintains or even enhances overall performance of various GNNs.
> > >
> > > We hope this provides clarity on the matter. If you have any further questions, we are happy to discuss.
> > >
> > > [1] Dong, Yushun, et al. "Fairness in graph mining: A survey." IEEE Transactions on Knowledge and Data Engineering (2023).
> > >
> > > Best,
> > >
> > > All authors

---

> > > > ### Comment · Reviewer_VK13 · 2023-08-18
> > > >
> > > > Dear Authors,
> > > >
> > > > I have raised my score due to the better clarification.

---

> > > > > ### Author Response · Authors · 2023-08-18
> > > > > **Thanks for your response**
> > > > >
> > > > > Dear Reviewer VK13,
> > > > >
> > > > > Thanks for your response and support. We are glad to know that our rebuttal has addressed your concerns.
> > > > >
> > > > > Best,
> > > > >
> > > > > All authors

---

### Author Rebuttal · Authors · 2023-08-10

# Global Response

We thank the reviewers for the valuable comments and suggestions. We carefully revise our paper and provide responses to address the concerns. In this global response, we are willing to provide information about tables and figures in the rebuttal pdf file and some common concerns.

* Figure 1 is designed to elucidate the label position bias across various models. We've selected a mix of five models for this demonstration: four GNNs—ReNode[1], GCNII[2], JKNet[3], and MAGNA[4]—that adhere to the message-passing scheme, and one graph transformer model, NodeFormer[5]. Our evaluations span both the Cora and CiteSeer datasets.
* Table 1 showcases the performance and label position bias when using the GCNII and JKNet backbones. We conduct experiments on both Cora and CiteSeer datasets.
* Table 2 illustrates how the performance and label position bias would be affected if using the proposed $l_1$-regularized Label Position Unbiased Sparse Structure Learning. We conduct experiments
    for the GCN and APPNP backbones, utilizing the learned graphs with varying sparsity levels on the Cora dataset. As sparsity intensifies, there's a slight dip in model performance. However, the label position bias remains consistent irrespective of the sparsity level.
* Figure 2 visualizes the results of significance tests conducted to confirm the statistical significance of the observed label position bias. For each experiment, we ran it 10 times and subsequently apply the t-test between every pair of groups. The outcomes reveal that most of the p-values are beneath the 0.05 threshold, a conventional benchmark for statistical significance. It's worth noting that only a minor fraction of adjacent groups did not adhere to this trend. These findings substantiate that the label position bias is statistically significant.
* Figure 3 is designed to explore the presence of label position bias in heterophily graphs. We observe a label position bias for the Chameleon dataset, but its pattern diverges from what we typically see in homophily graphs. Specifically, nodes in close proximity to labeled nodes tend to underperform, aligning with expectations based on the heterophily principle. Conversely, the Squirrel dataset doesn't exhibit a pronounced label position bias.
* Table 3 presents the performance and label position bias of our proposed method when faced with imbalanced class distributions. The results affirm that our method maintains its efficacy and performs commendably in such scenarios.

### **Common Concern 1: What is the root cause of the label position bias in GNNs?**

**Answer:** We attribute the main reason for label position bias to the inherent message passing in GNNs.

**Experimental evidence 1**: From Figure 1 in the rebuttal pdf file and Figure 1, 2, and 3 in our main paper, all 7 models with message passing, such as GCN, APPNP, GCNII, etc, all suffer from label position bias. However, the graph transformer-NodeFormer, which excludes message passing, has a small label position bias.

**Theoretical evidence 1**: The message passing intrinsically ties the representation of test nodes to labeled nodes. As established in [3], the influence score $I_i(j)$ can be employed to gauge the influence from node j to node i. [3] further shows that for a k-layer GCN, the influence distribution $I_x$ for any node corresponds to the k-step random walk distribution initiating from node x. Our conceptualized label proximal score is essentially a sum of the random walk probabilities from all labeled nodes directed toward the target node. An unfair graph structure can then produce inconsistent influences from labeled nodes to different test nodes.

**Experimental evidence 2:** As highlighted in Proposition 3.1 of our paper, by learning a fair graph structure, we can ensure uniform influence from training nodes to every test node. The experimental results in Table 2 of the main paper and Table 1 of the rebuttal pdf file demonstrate that this equitable influence significantly diminishes the label position bias in various GNNs, such as GCN, APPNP, GCNII, and JKNet.

Based on the above evidence, we believe that message passing with the unfair graph structure should be one of the main reasons.


### **Common Concern 2: More backbone models should be added to verify the proposed LPSL method.**

**Answer:**  We test our proposed LPSL with both the GCNII and JKNet backbones. The results in Table 1 of the rebuttal file indicate that our proposed LPSL method not only amplifies the performance of these backbone models but also effectively mitigates the label position bias.


[1] Chen, Deli, et al. "Topology-imbalance learning for semi-supervised node classification." NeurIPS 2021.

[2] Chen, Ming, et al. "Simple and deep graph convolutional networks." ICML 2020.

[3] Xu, Keyulu, et al. "Representation learning on graphs with jumping knowledge networks." ICML 2018.

[4] Wang, Guangtao, et al. "Multi-hop attention graph neural network."

[5] Wu, Qitian, et al. "Nodeformer: A scalable graph structure learning transformer for node classification." NeurIPS 2022.

---

### Decision · Program_Chairs · 2023-09-21

**Decision:**

Accept (poster)

**Comment:**

After the rebuttal and the added experiments, all reviewers voted for acceptance.
Please include the additional results as well as the proposed changes in the revised version of the manuscript.